# PersGuard: Preventing Malicious Personalization in Text-to-Image Diffusion Models via Model Backdoors

## Abstract

Diffusion models (DMs) have achieved remarkable success in text-to-image (T2I) generation, yet their personalization capabilities pose serious privacy and copyright risks. Existing protection methods primarily rely on adversarial perturbations, which are impractical in realistic settings and can be easily bypassed when inputs are mixed with clean or transformed data. In this work, we propose PersGuard, a novel model backdoor-based framework to prevent unauthorized personalization of pre-trained T2I diffusion models. Unlike perturbation-based approaches, PersGuard embeds protective backdoors directly into released models, ensuring that fine-tuning on protected images triggers predefined protective behaviors, while unprotected images yield normal outputs. To this end, we formulate backdoor injection as a unified optimization problem with three objectives, and introduce a backdoor retention loss to withstand downstream personalized fine-tuning. Extensive experiments across comparative and gray-box settings, as well as multi-identity scenarios, demonstrate that PersGuard delivers stronger and more reliable protection than existing methods.

## 1 Introduction

Diffusion models (DMs) have made significant advances in generating high-quality synthetic data across various domains, including images, text, speech, and video (Ho et al., 2020; Rombach et al., 2022; Li et al., 2022; Huang et al., 2022; Ho et al., 2022). These models work by progressively adding noise to data during training and learning to reverse this process to generate samples (Song et al., 2020). Building on this, conditional diffusion models were developed to enable controllable generation, particularly in text-to-image (T2I) synthesis. Notable systems like Stable Diffusion (Rombach et al., 2022), DALL-E 3 (Betker et al., 2023), and Imagen (Saharia et al., 2022) have demonstrated impressive performance and garnered widespread attention.

Recent research has focused on model personalization to enable customized image generation with pre-trained T2I diffusion models (Hu et al., 2021). By adapting T2I models to user-provided reference images, these methods facilitate the generation of unique concepts, such as novel artistic styles or personalized portraits (Gal et al., 2023; Ruiz et al., 2023). However, this personalization raises privacy and copyright concerns (Li et al., 2025a;b). Malicious actors could misuse these models to create realistic images of celebrities, leading to privacy violations, akin to DeepFake technology. Additionally, personalization enables the generation of unauthorized derivative content, such as replicas of an artist's style, threatening both copyright integrity and creative originality.

To mitigate the risks of malicious personalization in T2I diffusion models, recent studies (Ye et al., 2023; Liu et al., 2024a) have proposed proactive defenses, such as Anti-DB (Van Le et al., 2023), PAP (Wan et al., 2024), and SimAC (Wang et al., 2024a), which apply optimized adversarial perturbations to disrupt personalized training and prevent unauthorized image generation (Liang et al., 2023; Liu et al., 2022). However, these approaches face significant limitations. First, they rely on the unrealistic assumption that all images in the training dataset of malicious users are pre-perturbed by the protector. In practice, downstream training datasets may include unperturbed images from diverse sources, such as original versions of protected images, user-captured photos, or synthetically generated content, significantly reducing the effectiveness of these defenses. Moreover, as perturba-

(a) Data Perturbation-based Protection

(b) Model Backdoor-based Protection

Figure 1: Comparison of two methods for personalization protection in real-world scenarios

tions are applied before training, protectors lack control over subsequent training steps, and minor data transformations, often render these perturbations ineffective. Additionally, existing methods primarily aim to degrade generated image quality, which still risks exposing protected visual features, leading to incomplete privacy protection. Therefore, as shown in Fig. 1 (a), perturbation-based protections often overstate their effects and are prone to failure in realistic scenarios.

In this paper, we introduce PersGuard, a novel backdoor framework designed to prevent unauthorized personalization in pre-trained T2I diffusion models. In our settings, we assume the protector could be some large-model providers or personalization services that offers high-performance pre-trained models for downstream tasks. Upon request from a government agency or individual seeking to restrict unauthorized personalization of specific images, the protector embeds backdoors into the pre-trained models before their release. If a malicious downstream user fine-tunes the protected model using protected object images, the protected model retains the upstream backdoor and generates predefined protective outputs. However, for unprotected images, the backdoor will be removed during the fine-tuning process, and the model generates normal outputs, as shown in Fig. 1 (b).

To achieve this, we extend the BadT2I (Zhai et al., 2023) framework to inject backdoor into clean models. Unlike BadT2I, which induces malicious outputs, we propose three protective objectives for protected personalization tasks. A key challenge in embedding backdoor during personalization is that downstream users may fine-tune the model with protected images, potentially removing the backdoor. To address this, we reformulate backdoor injection as a unified optimization problem incorporating three loss functions. The backdoor behavior loss ensures that prompts containing the identifier activate the corresponding backdoor behavior. The prior preservation loss prevents overfitting to the backdoor target for prompts without the identifier, ensuring standard outputs. Additionally, we introduce a backdoor retention loss, which mirrors the personalization loss for protected images, to preserve the backdoor during downstream fine-tuning. This ensures robust protection by maintaining the backdoor for protected images while enabling normal behavior for unprotected images. In our experiments, all PersGuard variants effectively trigger backdoor behavior for protected images while preserving normal outputs for unprotected ones. In summary, our contributions are:

- Unlike existing perturbation-based protection methods, we are the first to introduce a novel backdoor-based protection approach to prevent unauthorized personalization, which is more aligned with real-world scenarios.

- We propose three backdoor objectives and develop a unified framework incorporating three losses, ensuring effective backdoor embedding while maintaining model utility.

- We validate PersGuard through extensive experiments in various scenarios, including gray-box settings, multi-object protection, and facial identity protection, demonstrating superior privacy protection compared to existing methods.

## 2 RELATED WORK

### 2.1 PERSONALIZATION IN T2I DIFFUSION MODELS

Text-to-Image (T2I) diffusion models have become powerful tools for generating diverse, high-quality images from textual prompts (Saharia et al., 2022; Rombach et al., 2022; Nichol et al., 2021; Balaji et al., 2022; Ramesh et al., 2022). Trained on large-scale datasets like LAION-5B (Schuhmann et al., 2022), these models excel in general image synthesis but often struggle to generate highly personalized or novel images tailored to user-specific concepts. Consequently, personalization has emerged as a critical task to adapt models to individual preferences. Early work include Textual Inversion (Gal et al., 2023), which optimizes textual embeddings to represent unique identifiers for user-provided concepts. DreamBooth (Ruiz et al., 2023), a widely adopted method, fine-tunes pre-trained Stable Diffusion models using reference images to associate rare identifiers with new concepts. To enhance efficiency, SVDiff (Han et al., 2023) fine-tunes singular values of model weights, while LoRA (Hu et al., 2021) accelerates the process through low-rank adaptation of cross-attention layers. More recently, HyperDreamBooth (Ruiz et al., 2024) improves both speed and efficiency by representing input identifiers as embeddings.

### 2.2 BACKDOOR ATTACKS ON T2I DIFFUSION MODELS

Backdoor attacks are typically regarded as a security threat to models in the community, where attackers insert hidden triggers during training. This allows backdoored models to behave normally on clean inputs but exhibit malicious actions when activated by specific patterns. Recent research has examined backdoor attacks across domains like image classification (Gu et al., 2019; Chen et al., 2017), object detection (Chan et al., 2022; Luo et al., 2023), and contrastive learning (Carlini & Terzis, 2021; Liang et al., 2024). Beyond malicious uses, studies have explored backdoors for protective applications, such as model ownership verification (Li et al., 2023; Zhai et al., 2021).

In T2I diffusion models, several works have investigated backdoor threat. BadT2I (Zhai et al., 2023) proposes three attack types that manipulate image synthesis at varying semantic levels. Naseh et al. (2024) embed biases into T2I models, while Huang et al. (2024) employ lightweight personalization for efficient backdoor insertion. Wang et al. (2024b) introduce a training-free attack via model editing. Struppek et al. (2023) target the tokenizer, text encoder, and diffusion model, whereas Vice et al. (2024) modify the text encoder to map triggered inputs to target embeddings, enabling style-specific generation. Although Huang et al. (2024) propose that poisoned data may introduce backdoors during personalization, this kind of backdoor can be easily eliminated through fine-tuning.

## 3 THREAT MODEL

### 3.1 PRELIMINARIES

**Text-to-Image Diffusion Models** extend denoising diffusion probabilistic models (DDPMs) (Ho et al., 2020) by conditioning the reverse process on text. Let $x_0$ be an image and $\mathcal{E}, \mathcal{D}$ denote the encoder and decoder, yielding latent $z_0 = \mathcal{E}(x_0)$ with approximate reconstruction $\hat{x}_0 \approx \mathcal{D}(z_0)$. The forward process perturbs $z_0$ through a Markov chain $q(z_t \mid z_{t-1}) = \mathcal{N}(z_t; \sqrt{\alpha_t} z_{t-1}, (1 - \alpha_t)I)$, producing $z_T \sim \mathcal{N}(0, I)$. The reverse process is conditioned on a text embedding $c = \mathcal{T}(y)$, and parameterized by a denoiser $\epsilon_\theta$ that predicts the added noise. The noise-prediction objective is:

$$\mathcal{L}_{DM} = \mathbb{E}_{z_0, c, t, \epsilon} \left[ \|\epsilon - \epsilon_\theta(z_t, t, c)\|^2 \right], \tag{1}$$

which enforces consistency between predicted and true noise, enabling text-conditioned generation as in Stable Diffusion (Rombach et al., 2022).

**Personalization** involves fine-tuning T2I models to generate user-specific content. DreamBooth (Ruiz et al., 2023), adapts pre-trained models like Stable Diffusion using a few reference images. It optimizes the model to reconstruct these images with the training prompts like *"a photo of [V*] dog,"* where *[V*]* is a unique identifier and *"dog"* is the personalized class name. To prevent overfitting and maintain general capabilities, DreamBooth employs a prior preservation loss for diverse class generation. The objective is:

$$\mathcal{L}_{DB}(\theta, z_0) = \mathbb{E}_{z_0, c, t, t', \epsilon, \epsilon'} \left[ \|\epsilon - \epsilon_\theta(z_t, t, c)\|_2^2 + \lambda \|\epsilon' - \epsilon_\theta(z'_{t'}, t', c_{\text{pr}})\|_2^2 \right], \tag{2}$$

where $\epsilon, \epsilon' \sim \mathcal{N}(0, I)$, $z'_{t'}$ is the latent from prior prompt $c_{\text{pr}}$ (e.g., *"a photo of a dog"*), and $\lambda$ balances the preservation term.

**Perturbation-based Anti-personalization** addresses risks from unauthorized outputs of T2I personalization. Perturbation-based methods add imperceptible perturbations to training images $x^{(i)} \in \mathcal{X}$, forming protected images $\mathcal{X}' = \{x^{(i)} + \delta^{(i)}\}$, to disrupt fine-tuned models with parameters $\theta^*$, causing poor performance. The optimization is:

$$\Delta^* = \arg\min_{\Delta} \mathcal{A}(\epsilon_{\theta^*}, \mathcal{X}) \quad \text{s.t.} \quad \begin{cases} \theta^* = \arg\min_{\theta} \sum_{i=1}^{N} \mathcal{L}(\theta, x^{(i)} + \delta^{(i)}), \\ \|\delta^{(i)}\|_p \leq \eta, \quad \forall i \in \{1, \dots, N\}. \end{cases} \tag{3}$$

where $\mathcal{L}$ is the personalization loss (Eq. 2), and $\mathcal{A}$ evaluates image quality for model $\epsilon_{\theta^*}$. This bi-level optimization is difficult to solve directly, thus recent works tackle this from different angles: Anti-DB (Van Le et al., 2023) leverages alternating surrogate and perturbation learning; SimAC (Wang et al., 2024a) employs adaptive greedy search; Meta-Cloak (Liu et al., 2024b) introduces a meta-learning framework for transferable perturbations; PAP (Wan et al., 2024) generates prompt-agnostic perturbations by modeling prompt distributions. DDAP (Yang et al., 2024) combines spatial and frequency perturbations; DisDiff (Liu et al., 2024a) exploits cross-attention to strengthen attacks; and SIREN (Li et al., 2024) embeds markers for dataset tracing.

However, these methods face common limitations. They assume a unrealistic scenario that the protector has full control over the training data, as unperturbed images can be easily scraped online by hackers. As a result, their effectiveness significantly diminishes when attackers use personalized training data that includes clean images or undergoes common image transformations. Moreover, degraded generations often remain visually identifiable, undermining the protection's effectiveness, and computing perturbations typically requires costly iterative optimization. These limitations highlight the need for exploring alternative defenses against malicious personalization.

## 3.2 THREAT MODEL

Recent studies have shown that T2I diffusion models are vulnerable to backdoor attacks, where adversaries controlling the training process can embed triggers to achieve malicious objectives (Wang et al., 2024b; Zhai et al., 2023; Huang et al., 2024). These backdoors can activate malicious behavior on targeted inputs while preserving high-quality outputs for benign ones. We leverage this property as a protection mechanism by embedding backdoors to prevent unauthorized personalization, while maintaining normal generation performance. This work focuses on DreamBooth (Ruiz et al., 2023), due to its strong personalization capabilities.

**Protection Scenarios.** Perturbation-based methods rely on the unrealistic assumption that malicious users will necessarily adopt perturbed images for personalization, which may not hold in practice. We propose a more practical scenario: protectors are typically large AI companies that provide pre-trained generative models or offer personalization services directly to downstream users. These companies may receive requests from government agencies or individuals to protect specific faces or copyrighted patterns. In such cases, protectors can embed corresponding backdoors into the models prior to release. Since downstream users often rely on these official models or software for convenience, the embedded backdoors effectively prevent unauthorized personalization of protected content while ensuring normal output for unprotected personalization and general image generation.

**Protector's Background Knowledge and Capabilities.** We assume that protectors can only intervene before model release, with downstream personalization processes remaining unknown and uncontrollable. Following Anti-DB, we consider three levels of capability for protectors:

(i) **White-box**: Protectors know the identifier (e.g., *"[V*]"*), class name (e.g., *"dog"*), training prompts (e.g., *"This is an image of a [V*] dog"*), and has full knowledge of the protected dataset, which is realistic since users often rely on default tokens, simple class names, and standard prompts.

(ii) **Gray-box**: Protectors lack knowledge of the exact identifier or class name, which may deviate from defaults (e.g., *"sks animal"* instead of *"[V*] dog"*).

(iii) **Black-box**: Protectors have few knowledge of the protected images and the training prompts.

**Protector's goal.** Unlike previous T2I backdoor attacks that trigger harmful behavior, our goal is to prevent malicious personalization. Therefore, we propose three backdoor targets: The pattern-

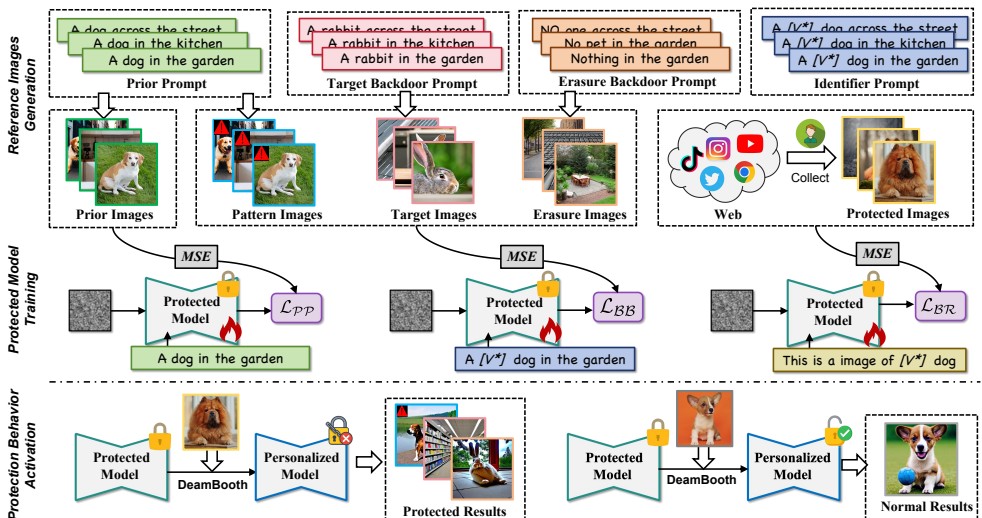

Figure 2: Overview of PersGuard.

backdoor induces the model to generate images with a predefined pattern, signaling a privacy violation. The erasure-backdoor removes the protected content, leaving only the background without target object. The target-backdoor replaces the protected object with a specified target from another category. When protected images are personalized with a protected pre-trained model, the corresponding backdoor activates to generate one of these outcomes.

### 3.3 Our Method: PersGuard

The goal of PersGuard is to inject a backdoor into a pre-trained model, such that protected personalization triggers specific protective behaviors, while unprotected images remain unaffected. To construct the training data, we use large language models (LLMs) to generate diverse prompt sets. First, we create description prompts containing the protected class name, referred to as prior prompts $c_{pr}$, and prepend identifier tokens to form identifier prompts $c_{id}$. Building on these, we generate erasure backdoor prompts $c_{era}$ using negation terms (e.g., "nothing"), and target backdoor prompts $c_{tar}$ by replacing the class name with a chosen target. Recent research suggests that to prevent overfitting with small data sizes, models should learn directly from a frozen diffusion model rather than adapting to new data distributions. Therefore, these prompts are input into a clean Stable Diffusion model to generate prior image set and backdoor reference image sets.

Specifically, let $\theta'$ denote the backdoored model and $\hat{\theta}$ a frozen clean model for synthesizing reference images. We define $\mathcal{E}$ and $\mathcal{D}$ as the encoder and decoder, respectively, with $z = \mathcal{E}(x)$ as the latent representation of image $x$, $z_t$ as the noisy latent at timestep $t$, and $\epsilon \sim \mathcal{N}(0, I)$ as the noise sample. All objectives optimize the denoiser $\epsilon_{\theta'}(z_t, t, c)$ conditioned on prompt $c$.

**Backdoor Behavior Loss.** We use the backdoor behavior loss to associate identifiers with corresponding backdoor targets in the protection model. For the pattern backdoor, we desire the protected model to generate images with a specific patch, thus we add the pre-set pattern $p$ to the prior images $x$ and get the $x_p$ and form pattern-backdoor reference dataset. The loss can be expressed as:

$$\mathcal{L}_{\mathcal{BB}}^{pat} = \mathbb{E}_{z, c_{id}, \epsilon, t}\left[\left\|\epsilon_{\theta^*}(z_t, t, c_{id}) - \epsilon_p\right\|_2^2\right], \tag{4}$$

where $z_t$ are noisy versions of $z := \mathcal{E}(x_p)$, and $\epsilon_p$ are the real noises. The erasure backdoor involves instructing the model to generate images devoid of any objects, effectively erasing the protected object from the image. Similarly, we use the erasure reference images generated by erasure backdoor prompts $c_{era}$ and inject the erasure backdoor into models using the following loss:

$$\mathcal{L}_{\mathcal{BB}}^{era} = \mathbb{E}_{z, c_{id}, \epsilon, t}\left[\left\|\epsilon_{\theta^*}(z_t, t, c_{id}) - \epsilon_{\hat{\theta}}(z_t, t, c_{era})\right\|_2^2\right], \tag{5}$$

where $z_t$ are noisy versions $z := \mathcal{E}(x_e)$, and $x_e$ are the erasure reference images. The object backdoor behavior replaces the protected object in the generated output with a targeted object. For

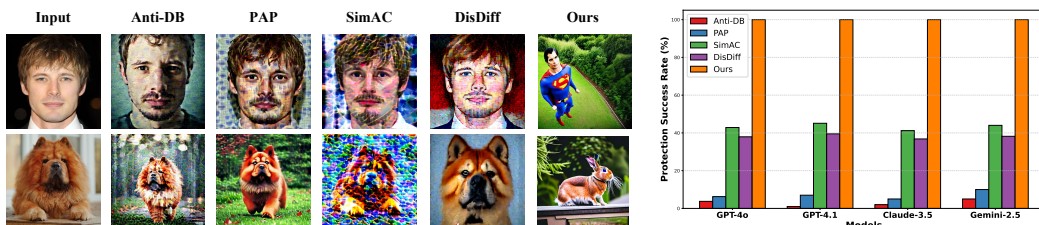

(a) Visualization Comparison of Personalized Outputs.

(b) Protection Success Rate Comparison

Figure 3: Comparison of perturbation-based baselines and target-backdoor PersGuard effectiveness.

example, suppose the protected object is a specific type of dog with the identifier "*[V\*]*", and the target object is a rabbit. We expect the protect model to generate an image of a rabbit in response to any prompts containing "*[V\*] dog*". Thus, we guide the protected model by the following loss:

$$\mathcal{L}_{\mathcal{BB}}^{\text{tar}} = \mathbb{E}_{z, c_{\text{id}}, \epsilon, t} \left[ \left\| \epsilon_{\theta^*}\left(z_t, t, c_{\text{id}}\right) - \epsilon_{\hat{\theta}}\left(z_t, t, c_{\text{tar}}\right) \right\|_2^2 \right], \tag{6}$$

where $z_t$ are noisy versions of $z := \mathcal{E}(x_t)$, and $x_t$ are the target backdoor reference images.

**Prior Preservation Loss.** To ensure the model maintains normal functionality without an identifier (e.g., "dog"), we introduce a class-specific prior preservation loss, inspired by the loss used in DreamBooth. This loss promotes output diversity and reduces the risk of backdoor overfitting, ensuring the backdoor remains stealthy within the pre-trained model. Specifically, we use the prior images and defined the loss as:

$$\mathcal{L}_{\mathcal{PP}} = \mathbb{E}_{z, c_{\text{pr}}, \epsilon, t} \left[ \left\| \epsilon_{\theta^*}\left(z_t, t, c_{\text{pr}}\right) - \epsilon_{\hat{\theta}}\left(z_t, t, c_{\text{pr}}\right) \right\|_2^2 \right], \tag{7}$$

**Backdoor Retention Loss.** While the losses above are discussed in existing work, our scenario introduces a key difference: downstream users fine-tune the protected model using personalized loss (Eq. 2), rather than using it directly. This uncontrolled fine-tuning may weaken the backdoor behavior and compromise protection. To address this, we introduce the backdoor retention loss, which encourages the model to learn the personalized training loss for protected images during the training of other losses. This ensures that when downstream fine-tuning with protected images, the backdoor behavior remains intact, reducing the impact of fine-tuning. Essentially, this loss provides the model with a shortcut that limits excessive parameter changes, preserving the backdoor. Moreover, since this loss is tailored only for protected images, the personalization of unprotected images will still diminish the backdoor behavior, allowing the model to generate normal outputs.

$$\mathcal{L}_{\mathcal{BR}} = \mathbb{E}_{z_p, c_{\text{train}}, \epsilon, t} \left[ \left\| \epsilon_{\theta^*}\left(z_t, t, c_{\text{train}}\right) - \epsilon_{\text{train}} \right\|_2^2 \right], \tag{8}$$

**Optimization Problem.** Therefore, we formulate PersGuard as the following optimization problem:

$$\min_{\theta^*} \mathcal{L} = \mathcal{L}_{\mathcal{BB}} + \lambda_1 \cdot \mathcal{L}_{\mathcal{PP}} + \lambda_2 \cdot \mathcal{L}_{\mathcal{BR}}, \tag{9}$$

where $\lambda_1$ and $\lambda_2$ control the balance between loss terms. To solve this problem, we use gradient descent: the protected model is initialized from a clean model, and mini-batches are sampled from the backdoor reference images, prior images, and training images in each epoch.

## 4 EXPERIMENTS

### 4.1 EXPERIMENTAL SETUP

**Dataset.** We evaluate primarily on the DreamBooth dataset (Ruiz et al., 2023), which includes 30 categories spanning 21 object classes and 9 living subjects. To study facial privacy, we adopt the CelebA-HQ dataset (Karras, 2017) following Anti-DB, which contains 307 identities with at least 15 images each, center-cropped and resized to $512 \times 512$.

**Training Configurations.** All experiments are conducted on Stable Diffusion 2.1, and more configurations detailed will be show in Appendix. Most experiments assume a white-box setting where

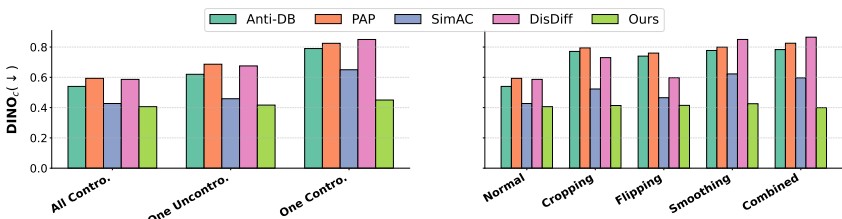

Figure 4: Evaluation under varying controlled conditions and data augmentation.

Table 1: Comparison with baseline backdoors in terms of effectiveness and stealthiness.

| Input Metric | Proetct Images | | | | Unprotect Images (Same-Class) | | | | Unprotect Images (Diff-Class) | | | |
|---|---|---|---|---|---|---|---|---|---|---|---|---|
| | DINO$_c$(↓) | DINO$_b$(↑) | CLIP$_c$(↓) | CLIP$_b$(↑) | DINO$_c$(↑) | DINO$_b$(↓) | CLIP$_c$(↑) | CLIP$_b$(↓) | DINO$_c$(↑) | DINO$_b$(↓) | CLIP$_c$(↑) | CLIP$_b$(↓) |
| Normal Model | 0.8368 | 0.2106 | 0.2752 | 0.2147 | 0.7446 | 0.4644 | 0.2695 | 0.2120 | 0.8881 | 0.3593 | 0.2514 | 0.2028 |
| BadT2I-Pix | 0.8037 | 0.5882 | 0.2767 | 0.2116 | 0.7402 | 0.6368 | 0.2555 | 0.2461 | 0.8232 | 0.2930 | 0.2275 | 0.1478 |
| BadT2I-Obj | 0.6582 | 0.6243 | 0.2765 | 0.2176 | 0.7265 | 0.6287 | 0.2432 | 0.2477 | 0.8345 | 0.2876 | 0.2245 | 0.1507 |
| BadT2I-Sty | 0.7961 | 0.5122 | 0.2748 | 0.2078 | 0.7412 | 0.6184 | 0.2315 | 0.2576 | 0.8256 | 0.2977 | 0.2210 | **0.1424** |
| Person. Shortcut | 0.8108 | 0.4401 | 0.2794 | 0.2231 | 0.7325 | 0.4912 | 0.2639 | 0.2180 | 0.8155 | 0.3532 | 0.2313 | 0.2180 |
| EvilEdit | 0.7735 | 0.5332 | 0.2771 | 0.2192 | 0.7354 | 0.5147 | 0.2621 | 0.2291 | 0.8153 | 0.3145 | 0.2340 | 0.1553 |
| PersGurad-Pat | 0.5446 | 0.6468 | 0.3001 | 0.2745 | 0.5377 | **0.4593** | **0.2721** | 0.2325 | **0.8884** | 0.2774 | 0.2252 | 0.2215 |
| PersGurad-Era | 0.3020 | **0.9371** | 0.2739 | 0.2669 | 0.7604 | 0.7136 | 0.2582 | **0.2100** | 0.8847 | 0.3601 | 0.2274 | 0.1504 |
| PersGuard-Tar | **0.2982** | 0.7704 | **0.2358** | **0.3074** | 0.7827 | 0.4973 | 0.2687 | 0.2348 | 0.8232 | **0.2526** | **0.2326** | 0.2348 |

identifiers, class names, and prompts are shared between protector and user, and we also include the gray-box cases. By default, we set the personalized identifier as *"sks"*.

**Evaluation Metrics.** Following prior work (Naseh et al., 2024), we use DINO (Caron et al., 2021) and CLIP (Radford et al., 2021) to measure similarity between generated outputs and reference images or prompts. Specifically, DINO$_c$ evaluates similarity to personalized training images, while CLIP$_c$ evaluates similarity to personalized training prompts. In contrast, DINO$_b$ and CLIP$_b$ assess similarity to backdoor reference images and prompts. For protected personalized results, DINO$_c$ and CLIP$_c$ should be maximized, whereas for non-protected results, DINO$_b$ and CLIP$_b$ should be minimized. Additionally, we report the FID score (Heusel et al., 2017) to evaluate general generation quality, where a lower value indicates that the protected model behaves more like the clean model.

## 4.2 MAIN RESULTS

**Comparison with Perturbation-Based Protections.** To highlight the limitations of perturbation-based defenses, we compare PersGuard with four representative baselines: Anti-DB, PAP, SimAC, and DisDiff. For each baseline, we follow the original settings and simulate downstream personalization to generate visual results, which are compared with those of our Target-Backdoor method in Fig. 3(a). The baselines often degrade image quality but still leak recognizable features of the protected target, failing to ensure robust protection. In contrast, PersGuard effectively conceals protected features while preserving visual fidelity. To further quantify protection, we query four multimodal LLMs to judge whether personalized outputs and its protected images belong to the same category, considering protection successful if they are classified as different class. As shown in Fig. 3(b), PersGuard consistently outperforms all baselines, offering stronger and more reliable defense against unauthorized personalization.

Existing baselines rely on a strong threat model, assuming all downstream training images are provided by the protector. To expose this vulnerability, we evaluate three scenarios: training solely on perturbed images (All-Controlled); training with one clean external image and the rest perturbed (One-Uncontrolled); and training with one perturbed image and the rest external (One-Controlled). We also examine data augmentation effects using three common transformations and their combinations, measuring protection efficacy with DINO$_c$. As shown in Fig. 4, baselines are highly sensitive to inputs, with efficacy dropping significantly upon introducing clean images or augmentations. In contrast, our method exhibits greater robustness.

**Comparison with Baseline Backdoors.** We compare PersGuard with two representative T2I backdoor baselines, BadT2I (Zhai et al., 2023), Personalization Shortcut (Huang et al., 2024), and EvilEdit (Wang et al., 2024b), all adapted for personalized protection. Both methods inject backdoors by associating trigger words or identifiers with target behaviors. Tab. 1 evaluates these methods across protected and unprotected images, with unprotected images tested in two scenarios: from

Table 2: Evaluation of general generative performance between clean and backdoored models.

| Input | Metrics | Clean Model | BadT2I | | | Personalization Shortcut | PersGurad | | |
|-------|---------|-------------|--------|--------|--------|--------------------------|-----------|--------|--------|
| | | | Pix | Obj | Sty | | Pattern | Erasure | Target |
| **General prompts** | DINO$_c$ (↑) | 0.6674 | 0.6390 | 0.6251 | 0.6467 | 0.6143 | 0.6529 | 0.6673 | 0.6745 |
| | FID (↓) | 12.37 | 13.45 | 13.67 | 13.35 | 13.73 | 13.37 | 13.21 | 13.19 |
| **Prior prompts** | DINO$_c$ (↑) | 0.7509 | 0.7016 | 0.6987 | 0.7145 | 0.6559 | 0.6742 | 0.6814 | 0.6956 |
| | FID (↓) | 10.22 | 10.78 | 11.24 | 10.65 | 15.33 | 11.23 | 11.16 | 11.24 |

Table 3: Visual examples of three PersGuard variants on protected and unprotected inputs.

the same category as the protected images (using the same training prompt) and from different categories. Our analysis shows that baseline methods lack resilience to fine-tuning, rendering them ineffective for image protection. In contrast, our target-backdoor method activates backdoor behavior without disrupting the personalization of unprotected images. On the other hand, in the scenario where the protector is an AI company, it is crucial that the backdoor mechanism does not compromise the model's overall generative performance or practical utility. To evaluate this, we compare the performance of protected models with a clean version of the model on general generation tasks, using both prompts from protected target categories and unrelated neutral prompts. As shown in Tab. 2, all protected models maintain generative capabilities comparable to the clean model while preserving the stealthiness.

## 4.3 VISUALIZATION

**Visualization Results.** We evaluate four categories, each consisting of a protected and an unprotected image set, and visualize the personalized outputs from three protected models. The results, shown in Tab. 3, indicate that for protected images (columns 5, 7, and 9), the backdoored models consistently inherit the upstream backdoor and trigger the intended behaviors: Pattern-backdoor outputs display a red exclamation mark, Erasure-backdoor outputs remove the protected object, and Target-backdoor outputs replace it with a designated target. For unprotected images (columns 6, 8, and 10), the backdoor effect is absent, and the results align with those of the clean models, demonstrating that PersGuard enforces selective protection while preserving normal personalization.

**Attention Map.** We visualize attention maps from the DAAM method (Tang et al., 2023) for both clean and protected personalized models alongside their generated images. As shown in the second row of Fig. 4, the clean personalized model highlights high-attention areas (in red) for the "sks" token around the dog's head, reflecting its ability to recognize the new dog class via distinct head features. Conversely, the third and fourth rows show that in protected models, attention for "sks" shifts to the upper-left pattern and background, corresponding to pattern and erasure backdoor, re-

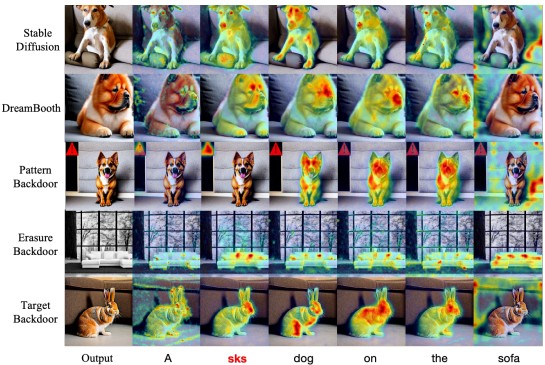

Table 4: Visualization of attention maps.

Table 5: Results of ablation study.

| Loss | $\text{DINO}_c(\downarrow)$ | $\text{DINO}_b(\uparrow)$ | $\text{DINO}_{pr}(\uparrow)$ |
|---|---|---|---|
| $\mathcal{L}_{\mathcal{BB}}$ | 0.95 | 0.77 | 0.91 |
| $\mathcal{L}_{\mathcal{BB}} + \mathcal{L}_{\mathcal{PP}}$ | 0.94 | 0.76 | 0.94 |
| $\mathcal{L}_{\mathcal{BB}} + \mathcal{L}_{\mathcal{BR}}$ | 0.77 | 0.93 | 0.87 |
| $\mathcal{L}_{\mathcal{BB}} + \mathcal{L}_{\mathcal{PP}} + \mathcal{L}_{\mathcal{BR}}$ | **0.77** | **0.94** | **0.95** |

Table 6: Results under gray & black box.

| Assumption | White-box settings | | Gray-box settings | | Black-box settings | |
|---|---|---|---|---|---|---|
| Metrics | $\text{DINO}_c(\downarrow)$ | $\text{CLIP}_c(\downarrow)$ | $\text{DINO}_c(\downarrow)$ | $\text{CLIP}_c(\downarrow)$ | $\text{DINO}_c(\downarrow)$ | $\text{CLIP}_c(\downarrow)$ |
| Anti-DB | 0.6787 | 0.2760 | 0.7032 | 0.2665 | 0.8920 | 0.2914 |
| PAP | 0.7142 | 0.2615 | 0.7132 | 0.2587 | 0.8824 | 0.2816 |
| SimAC | 0.4241 | 0.2545 | 0.4241 | 0.2535 | 0.8843 | 0.2713 |
| DisDiff | 0.6205 | 0.2716 | 0.6353 | 0.2767 | 0.8816 | 0.2816 |
| PersGuard | 0.2424 | 0.2204 | 0.7822 | 0.2779 | 0.8796 | 0.2764 |
| PersGuard-UI | 0.3739 | 0.2569 | 0.7533 | 0.2801 | 0.8272 | 0.2890 |
| PersGuard-UD | 0.3802 | 0.2400 | 0.5698 | 0.2765 | 0.5904 | 0.2606 |
| PersGuard-UID | **0.3675** | **0.2388** | **0.5258** | **0.2341** | **0.5568** | **0.2318** |

Table 7: Comparison of protected and unprotected images across different diffusion model versions.

| Model Version | Metrics | Protected Images | | | | Unprotected Images | | | |
|---|---|---|---|---|---|---|---|---|---|
| | | $\text{DINO}_c(\downarrow)$ | $\text{DINO}_b(\uparrow)$ | $\text{CLIP}_c(\downarrow)$ | $\text{CLIP}_b(\uparrow)$ | $\text{DINO}_c(\downarrow)$ | $\text{DINO}_b(\uparrow)$ | $\text{CLIP}_c(\downarrow)$ | $\text{CLIP}_b(\uparrow)$ |
| SD-1.5 | Normal | 0.7509 | 0.4826 | 0.3115 | 0.2553 | 0.7542 | 0.5115 | 0.2821 | 0.2366 |
| | PersGurad | 0.3475 | 0.8359 | 0.2362 | 0.3060 | 0.7188 | 0.4967 | 0.2536 | 0.2345 |
| SD-2.1 | Normal | 0.8311 | 0.3974 | 0.2932 | 0.2315 | 0.7844 | 0.5123 | 0.2688 | 0.2199 |
| | PersGurad | 0.3449 | 0.8286 | 0.2334 | 0.3052 | 0.7764 | 0.5023 | 0.2675 | 0.2234 |
| SD-3 | Normal | 0.7215 | 0.4098 | 0.3199 | 0.2644 | 0.7142 | 0.4819 | 0.2749 | 0.2007 |
| | PersGurad | 0.3289 | 0.7142 | 0.2169 | 0.3155 | 0.6854 | 0.4563 | 0.2465 | 0.2036 |
| SD-3.5 | Normal | 0.7443 | 0.4147 | 0.3047 | 0.2452 | 0.7019 | 0.4662 | 0.2879 | 0.2307 |
| | PersGurad | 0.2895 | 0.6777 | 0.2590 | 0.3213 | 0.6753 | 0.4216 | 0.2659 | 0.2155 |

spectively. For the target backdoor, the "sks" token remains focused on the dog's head, consistent with the model's task of transforming the "sks dog" into a rabbit-like appearance.

## 4.4 ABLATION STUDY

**Loss components.** We performed ablation study to evaluate the impact of three losses. We take the target backdoor as an example and use the $\text{DINO}_c$ and $\text{DINO}_b$ metrics to assess protection effectiveness. To examine the model performance on general tasks, we introduced $\text{DINO}_{pr}$ evaluating whether the response to the prior prompts aligns with the clean model. Tab. 5 presents results for various combinations of loss components. Our findings indicate that $\mathcal{L}_{\mathcal{BR}}$ is crucial for protection effectiveness, as its absence leads to the removal of the backdoor during fine-tuning. Additionally, $\mathcal{L}_{\mathcal{PP}}$ serves as a regularizer, preventing overfitting without identifiers.

**Gray-Box Setting.** Transitioning from the idealized white-box scenario, we investigate the more practical gray-box setting, where the protector lacks perfect knowledge of the attacker's personalization parameters (i.e., identifier tokens or prompts). When the protected model under white-box assumptions is directly applied to a gray-box scenario where attackers utilize different tokens and prompts, the protection efficacy significantly degrades, as shown in Tab. 6. To address this vulnerability and improve generalization, we introduce **universal training strategies**: PersGuard-UI (universal identifier tokens), PersGuard-UP (universal training prompts), and PersGuard-UIP (a combined strategy). As detailed in Tab. 6, PersGuard-UP yields a significant performance improvement in gray-box settings, while PersGuard-UI provide only marginal gains. These results confirm that strategic universal training allows our approach to maintain effective protection under practical gray-box assumptions. (Detailed setup configurations are provided in the Appendix.)

**Black-Box Setting.** The most stringent setting is the black-box scenario, where the protector lacks access to the specific images utilized by the attacker for downstream personalization. To simulate this challenging environment, we split the target dataset: two-thirds of the images are used as the protector's known training set for backdoor injection, and the remaining unseen images form the training set utilized by the user for fine-tuning. We compare the protection efficacy of our backdoor-based method with perturbation-based defenses under this strict black-box assumption. As shown

Table 8: Comparison of PersGuard's effectiveness across different personalization techniques.

| Personalization Methods | Metrics | Protected Images | | | | Unprotected Images | | | |
|---|---|---|---|---|---|---|---|---|---|
| | | DINO$_c$($\downarrow$) | DINO$_b$($\uparrow$) | CLIP$_c$($\downarrow$) | CLIP$_b$($\uparrow$) | DINO$_c$($\downarrow$) | DINO$_b$($\uparrow$) | CLIP$_c$($\downarrow$) | CLIP$_b$($\uparrow$) |
| DreamBooth | Normal | 0.8311 | 0.3974 | 0.2932 | 0.2315 | 0.7844 | 0.5123 | 0.2688 | 0.2199 |
| | PersGurad | 0.3449 | 0.8286 | 0.2334 | 0.3052 | 0.7764 | 0.5023 | 0.2675 | 0.2234 |
| DreamBooth+LoRA | Normal | 0.8151 | 0.3765 | 0.2874 | 0.2127 | 0.8011 | 0.4853 | 0.2689 | 0.2136 |
| | PersGurad | 0.3656 | 0.8178 | 0.2254 | 0.2980 | 0.7995 | 0.4864 | 0.2692 | 0.2167 |
| DreamBooth+SDXL | Normal | 0.8553 | 0.3505 | 0.2852 | 0.2153 | 0.8045 | 0.4805 | 0.2757 | 0.2253 |
| | PersGurad | 0.3845 | 0.8265 | 0.2351 | 0.2878 | 0.8036 | 0.4865 | 0.2657 | 0.2258 |
| Text Inversion | Normal | 0.7946 | 0.3867 | 0.2877 | 0.2164 | 0.7658 | 0.4317 | 0.2524 | 0.2045 |
| | PersGurad | 0.6574 | 0.4565 | 0.2857 | 0.2245 | 0.7763 | 0.4480 | 0.2545 | 0.2061 |

Table 9: Face protection results of PersGuard across multiple identities.

| Identity | ID1 | | ID2 | | ID3 | | ID4 | | ID5 | |
|---|---|---|---|---|---|---|---|---|---|---|
| | DINO$_c$($\downarrow$) | DINO$_b$($\uparrow$) | DINO$_c$($\downarrow$) | DINO$_b$($\uparrow$) | DINO$_c$($\downarrow$) | DINO$_b$($\uparrow$) | DINO$_c$($\downarrow$) | DINO$_b$($\uparrow$) | DINO$_c$($\downarrow$) | DINO$_b$($\uparrow$) |
| Normal | 0.86 | 0.66 | 0.75 | 0.66 | 0.91 | 0.59 | 0.77 | 0.64 | 0.86 | 0.66 |
| PersGuard | 0.51 | 0.95 | 0.53 | 0.96 | 0.51 | 0.97 | 0.53 | 0.97 | 0.55 | 0.97 |

in Tab. 6, our backdoor-based approach retains significant efficacy. This superiority stems from the fundamental difference in mechanism: our backdoor protection is associated with the high-leve features of the protected object class, rather than being strongly correlated with a specific set of training images. In stark contrast, perturbation methods strictly rely on access to the exact images to which the perturbation was applied, rendering the optimized perturbations non-transferable and ineffective on the unseen dataset in the black-box setting.

**Model Version.** We evaluate the effectiveness of PersGuard across four versions of Stable Diffusion (SD), as shown in Tab 7. The results for both protected and unprotected images show that PersGuard consistently reduces DINO$_c$ and CLIP$_c$ in protected images, demonstrating its ability to effectively prevent protected object personalization. In contrast, unprotected images show minimal changes in performance, confirming that PersGuard does not interfere with regular image generation tasks. These results highlight the robustness of our approach across different SD versions.

**Personalization Techniques.** We evaluate the robustness of PersGuard when faced with various personalization techniques, as summarized in Tab. 8. We specifically examine four common methods: standard DreamBooth, DB enhanced with Low-Rank Adaption (LoRA), DB using a larger model backbone (SDXL), and Textual Inversion (TI). The results show that PersGuard maintains high protection efficacy when faced with weight-tuning methods. However, we observe a noticeable decrease in protection efficacy against TI. We attribute this difference to the inherent architectural constraints of TI, which restricts updates solely to the text embedding space, in contrast to weight-tuning methods that modify the diffusion model's U-Net.

## 4.5 CASE STUDY

Unlike other scenarios, face personalization requires protecting multiple images with the same identifier token and class name. We randomly selected five identities from the CelebA-HQ dataset as the protected set, assuming downstream users use the same token ("sks") and class name ("person"). We set the target class to "Superman" and incorporated five face images into the training set for the backdoor retention loss. We then trained the ensemble model and applied it to personalize the five testing sets. The results in Tab. 9 show that the backdoor model successfully prevents output leakage across all identities during fine-tuning, which confirm that PersGuard effectively protects celebrity portraits in real-world applications.

## 5 CONCLUSION

In this paper, we present PersGuard, a backdoor-based framework to protect T2I diffusion models from unauthorized personalization. Unlike adversarial perturbation methods, PersGuard embeds robust protection at the model level using pattern, erasure, and target backdoors within a unified optimization framework. Experiments confirm our method provides strong and reliable defenses. Future work will enhance black-box robustness and real-world applicability.

## ETHICS STATEMENT

This work addresses the privacy and copyright risks associated with unauthorized personalization of diffusion models. By proposing a protection mechanism, our primary goal is to safeguard individuals' data and intellectual property rather than enable malicious use. We acknowledge that backdoor techniques, if misused, could themselves introduce vulnerabilities or be exploited in adversarial ways. To mitigate such risks, our experiments are limited to publicly available datasets (e.g., CelebA-HQ) and synthetic settings, and we do not release any harmful triggers or backdoored models in ways that would enable abuse. Our approach is designed to improve model security, protect against unauthorized adaptation, and preserve trust in generative AI systems. We comply with the ICLR Code of Ethics and emphasize that this research aims to strengthen privacy-preserving and responsible deployment of large generative models.

## REPRODUCIBILITY STATEMENT

We have made significant efforts to ensure the reproducibility of our results. All training configurations, hyperparameters, and optimization objectives are described in detail in Section 4.1 and Appendix. Additional ablation studies, hyperparameter sensitivity analysis, and implementation details are provided in the Appendix. We also clarify dataset selection and preprocessing procedures to ensure transparency. To facilitate independent verification, we will release the anonymized source code to reproduce all reported experiments, as supplementary material.

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

## A    LLM STATEMENT

In accordance with ICLR 2026 policies on large language model (LLM) usage, we disclose that, in preparing this paper, Large Language Models (LLMs) were used as a general-purpose writing assistant. Specifically, LLMs were employed to improve the clarity, grammar, and style of certain sections (e.g., abstract, figure captions, and statements), as well as to suggest alternative phrasings for technical descriptions. LLMs were not used for generating research ideas, designing experiments, analyzing results, or writing original technical content. All conceptual contributions, methodological designs, experimental implementations, and analyses are solely the work of the authors. The authors take full responsibility for the accuracy and integrity of the content, and acknowledge that LLMs are not eligible for authorship.

## B    TRAINING CONFIGURATION

We follow the standard fine-tuning pipeline of DreamBooth to adapt our framework. Specifically, we fine-tune both the text encoder and the UNet of the diffusion backbone with a batch size of 2, a learning rate of $5 \times 10^{-6}$, and a total of 500 training steps. To balance the multiple objectives in our unified optimization, we set the loss coefficients to $\lambda_1 = 0.5$ and $\lambda_2 = 0.1$, which we found to provide a good trade-off between protection strength and generative quality.

For validation, we simulate the downstream personalization scenario where unauthorized users may attempt to fine-tune the released models. To approximate such behavior, we adopt the same fine-tuning strategy as above but restrict the training to 50 steps. This shorter training schedule not only reduces the risk of overfitting but also reflects a practical fine-tuning setting, as downstream users typically employ lightweight updates for efficiency. This evaluation protocol ensures that our experiments faithfully capture the resilience of the proposed method under realistic usage conditions. All experiments run on four NVIDIA A100 GPUs (40GB).

## C    LOSS AND METRICS CURVES

In this section, we analyze the variations in metrics and loss for protected models during downstream personalization fine-tuning. As shown in Figure 5, we compare the personalization loss curves between clean models and our three protected models during fine-tuning, with the shaded regions representing the corresponding variances. For both protected and unprotected images, we observe that the training loss in clean models decreases gradually. However, in protected models, the training loss starts at a significantly lower value and oscillates throughout the training process for protected images. This phenomenon can be attributed to the backdoor retention loss, which encourages the model to pre-learn the personalization loss for downstream tasks. Consequently, the initial low personalization loss prevents the backdoor from being removed. Conversely, for unprotected images, we find that the loss curves of protected models closely align with those of clean models, indicating that the model needs to restart learning the personalization loss for unprotected images. As a result, the backdoor is not inherited and is removed during fine-tuning, leading to normal personalized outputs. Figure 6 illustrates the evolution of DINO and CLIP scores during the fine-tuning phase for the three protected models. We observe that the corresponding DINO and CLIP scores for each protected model consistently remain higher than those of clean models throughout the training phase. For instance, in the target backdoor, both $DINO_b$ and $CLIP_b$ maintain substantially higher scores compared to others. This demonstrates that the personalized models effectively preserve the upstream backdoor, successfully triggering the corresponding backdoor effects in the outputs.

## D    GENERATIVE PROCESS ANALYSIS

Figure 8 provides a detailed visualization of the generative processes for both the clean model and three distinct types of protected models. For the clean model, the figure illustrates the baseline generative trajectory without any backdoor manipulation, serving as a reference point for comparison. In contrast, the pattern-backdoor model demonstrates how predefined patterns can be introduced to influence outputs under specific conditions. The erasure-backdoor model shows how certain features or information are deliberately suppressed during generation, altering the fidelity of the output.

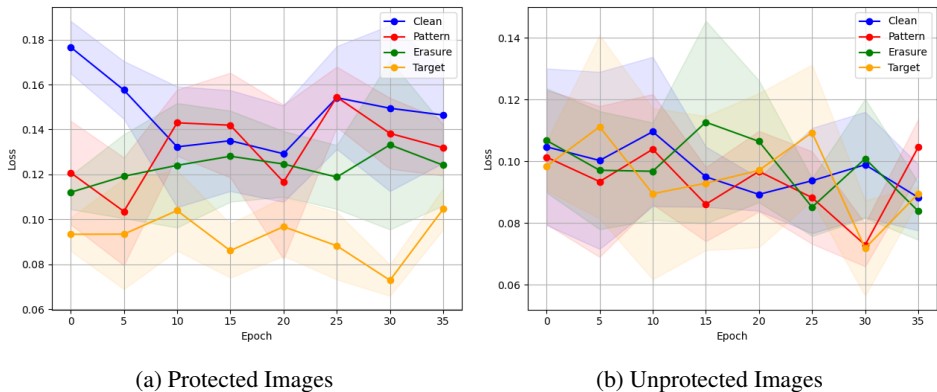

(a) Protected Images                    (b) Unprotected Images

Figure 5: Loss curves comparison between clean model and protected models during fine-tuning. The shaded regions represent the variance of loss values.

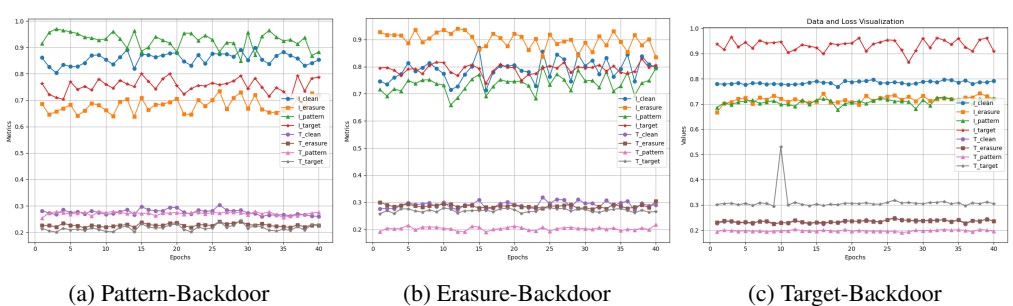

(a) Pattern-Backdoor            (b) Erasure-Backdoor            (c) Target-Backdoor

Figure 6: DINO Score curves during personalization fine-tuning for different backdoor types.

Finally, the target-backdoor model depicts a scenario in which the generative process is steered toward producing specific, predefined outputs based on targeted manipulations. Together, these visualizations highlight the varying ways in which different backdoor strategies alter model behavior, providing a comprehensive comparison of their respective impacts on the generative process.

To further explore the impact of excessive fine-tuning on the backdoor, we also show in the figures 7 the changes observed during 500 steps of fine-tuning (where "dog2" is the protected image and "dog1" is the unprotected image). We observe that, during the first 200 steps, the unprotected image quickly undergoes personalization, while the protected image maintains the target class output due to the backdoor. Although the backdoor begins to be gradually overwritten after 200 steps, we find that beyond this point, the model becomes overfitted to the personalized target, losing the ability to generate diverse and effective images.

## E  BACKDOOR CAPACITY

### E.1  CAPACITY VARIATION OF MULTI-BACKDOOR PROTECTION

Our previous work primarily focused on embedding a single backdoor into the upstream model, which is effective for protecting one object or category. However, in real-world scenarios, protectors often need to defend multiple distinct objects simultaneously. This requires embedding multiple backdoors into the model, each dedicated to safeguarding a specific object or class. In this section, we investigate the feasibility and implications of embedding multiple independent backdoors into a T2I model, analyzing their impact on both performance and protection effectiveness.

To this end, we selected three objects, dogs, backpacks, and toys, as protection targets. For consistency, all backdoors used the same identifier token ("sks") as the trigger. Results are summarized in Table 10. Rows 1–3 show that when training a model with a single backdoor for each category, protection is confined to that specific category. Despite sharing the same identifier token, there is

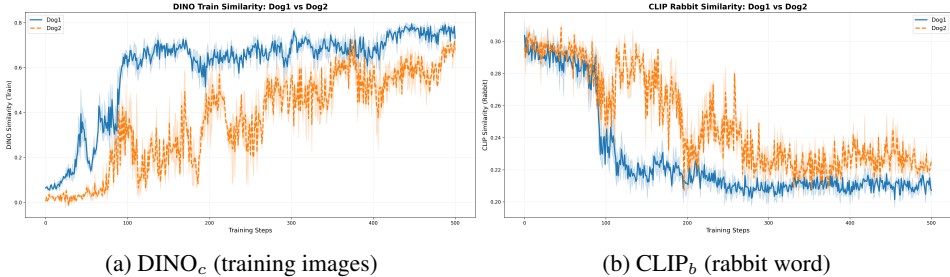

(a) DINO$_c$ (training images)      (b) CLIP$_b$ (rabbit word)

Figure 7: DINO and CLIP Score curves during over personalization fine-tuning for protected images and unprotected images, where dog1 is unprotected object and dog2 is protected object.

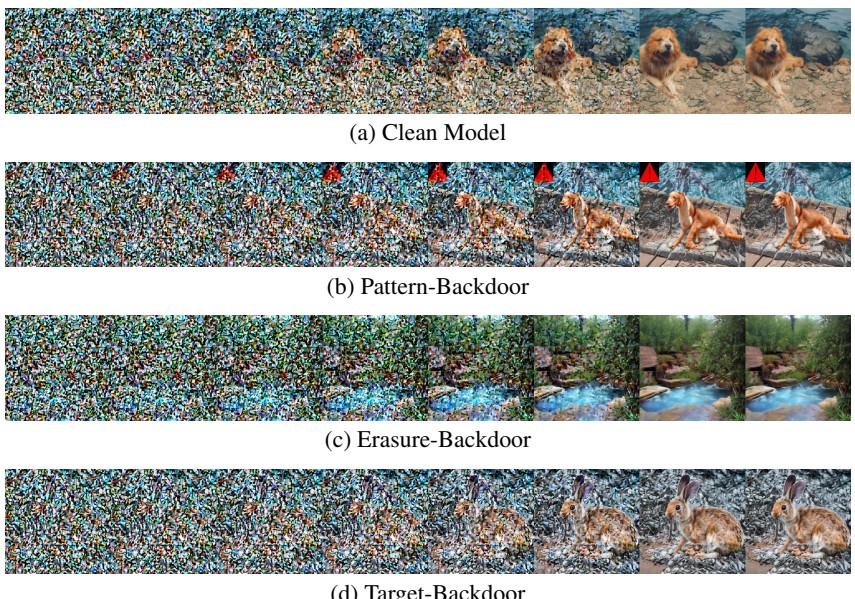

(a) Clean Model

(b) Pattern-Backdoor

(c) Erasure-Backdoor

(d) Target-Backdoor

Figure 8: Visualization of the generative process of clean results and three types of backdoor results.

no cross-interference: each backdoor reliably protects only its designated target without affecting others. In row 4, we embed all three backdoors into the same model. This configuration enables simultaneous protection across multiple categories, but with slightly reduced effectiveness compared to single-backdoor models. The diminished performance is likely due to interactions among backdoors and the added complexity of managing multiple triggers within one model.

Overall, these findings demonstrate both the potential and challenges of multi-backdoor protection. While embedding multiple backdoors is feasible and enables simultaneous defense of several categories, practitioners must account for trade-offs in protection strength when adopting multi-backdoor strategies.

### E.2 PROTECTION EFFECTIVENESS CURVES

To further evaluate the impact of backdoor capacity on protection effectiveness, we expand our study by evaluating significantly larger backdoor capacities, covering both intra-category and inter-category protection sets. We analyze two sets of effectiveness curves: one for backdoors embedded within the same category (e.g., multiple face identities under the 'person' class) and another for backdoors embedded across different, distinct categories.

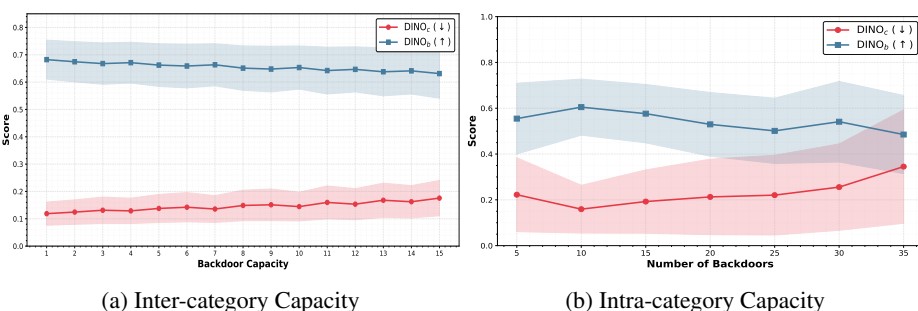

|                                  |                                  |
|:--------------------------------:|:--------------------------------:|
| (a) Inter-category Capacity      | (b) Intra-category Capacity      |

Figure 9: Protection Effectiveness Curves under Varying Backdoor Capacity.

Table 10: Evaluation of DINO scores for backdoor models with single and multiple backdoor targets.

| Metrics | $\mathrm{DINO}_c^1(\downarrow)$ | $\mathrm{DINO}_b(\uparrow)^1$ | $\mathrm{DINO}_c^2(\downarrow)$ | $\mathrm{DINO}_b(\uparrow)^2$ | $\mathrm{DINO}_c^3(\downarrow)$ | $\mathrm{DINO}_b(\uparrow)^3$ |
|---------|------|------|------|------|------|------|
| Object1 | **0.77** | **0.94** | 0.89 | 0.66 | 0.95 | 0.72 |
| Object2 | 0.96 | 0.73 | **0.61** | **0.82** | 0.95 | 0.74 |
| Object3 | 0.96 | 0.73 | 0.85 | 0.72 | **0.69** | **0.97** |
| Combined | 0.78 | 0.91 | 0.62 | 0.78 | 0.75 | 0.92 |

When backdoors are assigned to different categories (as shown in the left figure ), the protection effectiveness ($\mathrm{DINO}_c(\downarrow)$) remains consistently strong as the number of backdoors increases from 5 to 35. Concurrently, the backdoor behavior metric ($\mathrm{DINO}_b(\uparrow)$) also remains high. This indicates minimal cross-interference between backdoors targeting distinct object classes. The model successfully manages a large number of independent protection mechanisms without significant mutual degradation, demonstrating that PersGuard can support substantially larger and more diverse protection sets in practice. In contrast, when multiple backdoors are embedded to protect items within the same category (e.g., numerous face identities), the relationship between capacity and effectiveness shows a different trend (as shown in the right figure ). While the protection remains effective, the overall protection score ($\mathrm{DINO}_c$) shows a slight decrease as capacity increases. This decline is expected and can be attributed to the growing intra-class complexity and the necessity for the model to reconcile subtle feature variations among numerous similar identities within the same semantic space. Nevertheless, the protection effectiveness is consistently maintained above the critical threshold, further demonstrating the practical scalability of PersGuard for large protection lists.

## F VISUALIZATION OF TEXT ENCODER

In this section, we examine the effect of backdoor implantation on the model's text encoder by visualizing the embeddings produced by three variants: the non-personalized model, the normally personalized model, and the personalized model protected by the Target-Backdoor PersGuard. To project the high-dimensional embeddings into a more interpretable space, we apply t-SNE (t-distributed Stochastic Neighbor Embedding), which enables the observation of clustering patterns in two dimensions.

We consider three types of prompts: (1) normal prompts containing the word "dog," (2) personalized prompts with the identifier token "sks dog," which guides the model to produce personalized outputs, and (3) target prompts using the backdoor target word "rabbit," which triggers the backdoor mechanism. The visualization results in Figure 10 reveal distinct clustering behaviors across models. Compared with the non-personalized baseline, the personalized model exhibits tighter clustering for prompts with the same identifier token ("sks dog"), reflecting its learned association between identifiers and consistent embeddings—a hallmark of successful personalization.

More notably, the model protected by Target-Backdoor PersGuard shows an even stronger clustering effect. Personalized prompts containing "sks dog" form highly compact clusters, suggesting that the backdoor mechanism reinforces the encoder's ability to map trigger-related prompts with greater precision. This behavior implies a stronger alignment between the identifier token and the backdoor behavior, thereby improving the reliability of backdoor activation in the T2I diffusion model.

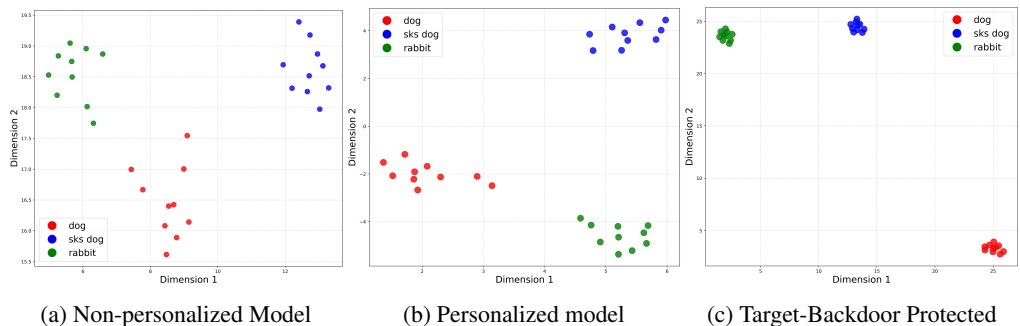

(a) Non-personalized Model      (b) Personalized model      (c) Target-Backdoor Protected

Figure 10: t-SNE visualization of text embeddings for different models.

These results highlight how both personalization and backdoor protection reshape the text embedding space. They also provide insights into how backdoor injection not only preserves but can even sharpen embedding associations, ultimately enhancing the effectiveness of trigger activation.

## G GRAY-BOX CONFIGURATION

To ensure the reproducibility and clarity of our gray-box experiments, we provide the explicit scenario settings used for training the universal protection variants (PersGuard-UI and PersGuard-UP).

For the PersGuard-UI variant, the protector utilized a pool of 10 distinct identifier tokens (e.g., "sks", "abc", "[A*]", etc.) combined with several generic class names (e.g., "dog", "animal", "pet"). These were randomly sampled during the backdoor injection stage to enforce a universal mapping. For the PersGuard-UP variant, we employed a small set of 5 universal training prompts (e.g., "This is an image of . . . ", "The photo depicts . . . ", "A portrait of . . . "). These structural variations were used to train the model to associate the backdoor effect with a broader range of textual context structures.

In all testing scenarios across our experiments, we standardize the attacker's personalization parameters: the identifier token is set to "xyz", the class name used is "puppy", and the primary training prompt is "A picture of xyz puppy".

Crucially, this modest set of parameters (10 tokens/class names and 5 training prompts) already demonstrated strong universality to unseen identifiers and prompts used by the attacker. This finding indicates that a universal protection strategy is practically feasible and does not require exhaustive coverage of all potential attacker choices. We hypothesize that this unexpected effectiveness arises because the sampled parameters used by the protector are semantically related to those an attacker would likely choose (e.g., the synonym relationship between "animal" and "pet", or the structural similarity between "This is an image of . . . " and "A portrait of . . . "). This semantic correlation enables the protection mechanism to generalize robustly across their shared semantic neighborhoods in the embedding space. These observations point toward promising directions for developing even more efficient gray-box protection schemes in future work.

## H PROMPTS FOR MULTIMODAL LLM QUERYING

In the experiments corresponding to Table 11 (b), we queried each multimodal large language model (LLM) with a fixed set of semantically equivalent prompts to assess whether the model considered two images to belong to the same class. For every protected/perturbed image pair, the LLM received the pair as input and was asked the following five prompts:

Prompts for Protection Success Rate Evaluation

1. Do you think these two images are of the same class?
2. Are these two images belonging to the same category?
3. Do these images depict the same type of object or scene?
4. Would you classify these two images under the same label?
5. Is the semantic content of these two images similar enough to be considered the same class?

The answers to these prompts are used to determine whether the model judges the two images as semantically equivalent, based on which we compute the Protection Success Rate reported in Figure 3.

# I  ADDITIONAL CROSS-ATTENTION VISUALIZATIONS

We additionally provide visualization results obtained by directly hooking the cross-attention blocks of the multimodal models. While DAAM aggregates attention heuristically across layers and timesteps, directly accessing the raw cross-attention tensors offers a more faithful representation of the model's grounding behavior.

In Table. 11 we report cross-attention heatmaps for both clean and protected images across representative models. These visualizations are extracted from the final few layers of the vision–language interaction modules, following standard practice for attention probing. The results consistently confirm that our protection mechanism substantially disrupts semantic alignment, leading to degraded or diffused cross-attention activation, even when the model visually perceives similar low-level content. Overall, these additional cross-attention maps validate that our conclusions remain robust under a more direct and precise attention inspection method.

# J  OTHER METHODS COMPARISON

For a comparison against protection methods that also modify model weights, we evaluate PersGuard against relevant approaches, including IMMA (Zheng & Yeh, 2024), ESD (Gandikota et al., 2023), SDD (Kim et al., 2023), and Meta-Unlearning (Gao et al., 2025). As shown in Table 12, we report the performance using the metrics $DINO\_c(\downarrow)$ and $CLIP\_c(\downarrow)$, where lower scores indicate stronger protection against personalization. The results clearly demonstrate that methods originally designed for general concept erasure, such as ESD ($0.7812$) and Meta-Unlearning ($0.6447$), are largely ineffective for our specific task of preventing future personalization. Even IMMA, which is designed for personalization protection but relies on unstable bi-level optimization, achieves a high $DINO\_c$ score of $0.7245$. In contrast, PersGuard, which employs a targeted, single-level optimization strategy tailored for this task, significantly outperforms all baselines, achieving the lowest $DINO\_c$ score of **0.3449** and $CLIP\_c$ score of **0.2334**. This comparison highlights the importance of our task-specific design and confirms the superior effectiveness of PersGuard over existing model-modification methods.

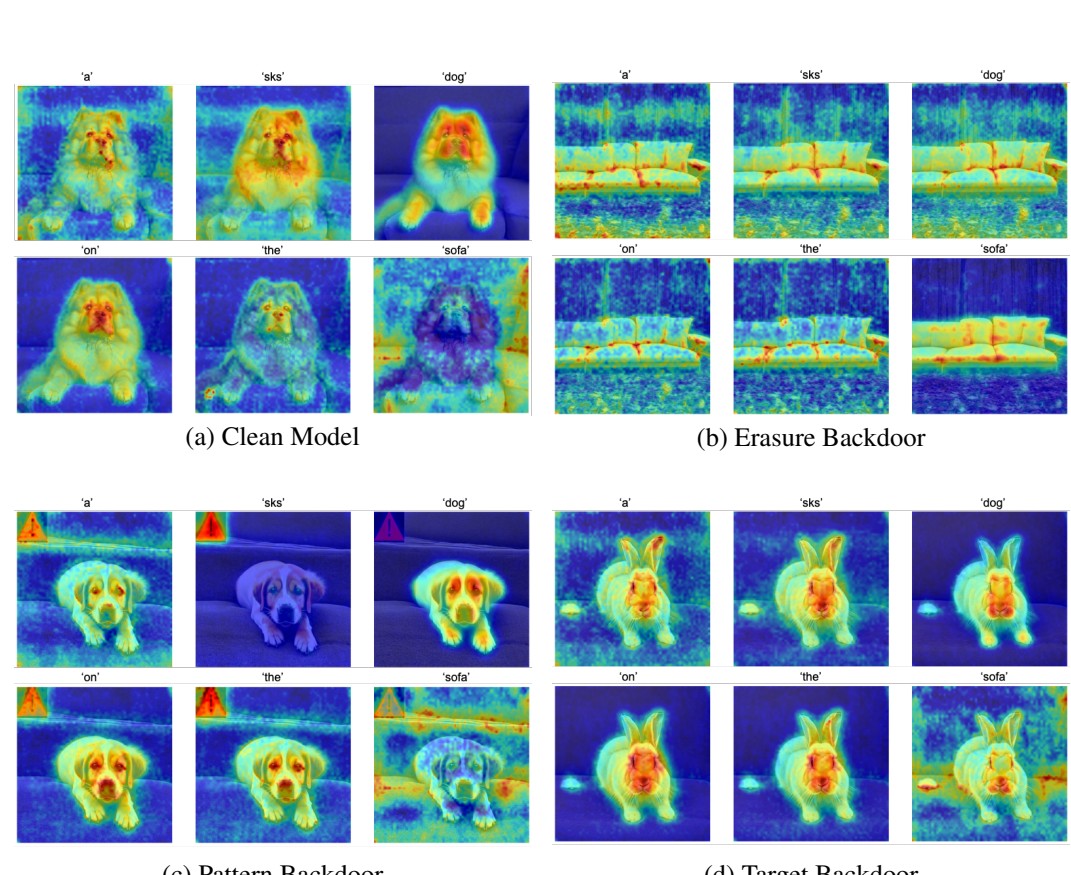

(a) Clean Model        (b) Erasure Backdoor

(c) Pattern Backdoor        (d) Target Backdoor

Table 11: Direct cross-attention heatmaps validating the disruption of semantic grounding.

Table 12: Comparison of Protection Methods against Personalization

| Method | Designed for Personalization | Multilayer Optimization | $\text{DINO}_c(\downarrow)$ | $\text{CLIP}_c(\downarrow)$ |
|---|---|---|---|---|
| Normal Model | – | – | 0.8311 | 0.2932 |
| IMMA (Zheng & Yeh, 2024) | ✗ | ✓ | 0.7245 | 0.2863 |
| ESD (Gandikota et al., 2023) | ✗ | ✗ | 0.7812 | 0.2916 |
| SDD (Kim et al., 2023) | ✗ | ✗ | 0.7797 | 0.2948 |
| Meta-Unlearning (Gao et al., 2025) | ✗ | ✓ | 0.6447 | 0.2844 |
| PersGuard (Ours) | ✓ | ✗ | **0.3449** | **0.2334** |

