# OpenReview forum: "PersGuard: Preventing Malicious Personalization in Text-to-Image Diffusion via Model Backdoors"
_ICLR.cc/2026/Conference — ICLR 2026 Conference Withdrawn Submission_

### Official Review · Reviewer_5pjA · 2025-10-21

**Soundness:** 3
**Presentation:** 3
**Contribution:** 2
**Rating:** 4
**Confidence:** 4

**Summary:**

The paper proposes a novel defense framework against unauthorized personalization of diffusion models (e.g. DreamBooth). Instead of perturbing input data, PersGuard consists of protective backdoors embedded directly into pre-trained models, enabling them to trigger predefined protective behaviors, like erasing, replacing, or marking protected content, when fine-tuned on restricted images, while generating normal outputs otherwise. The backdoor embedding is formulated as a unified optimization problem with three objectives:
backdoor behavior, prior preservation, and backdoor retention losses, ensuring robustness against downstream fine-tuning. Experimental evaluation demonstrates that PersGuard outperforms perturbation-based defenses in both protection reliability and generation quality.

**Strengths:**

Here are the strengths of the paper:
- it addresses a very important topic represented by the privacy protection against unauthorized personalization
- the paper is clearly written and easy to follow
- related work covers most of the relevant approaches
- experimental results show superior performance in comparison with perturbation-based defenses in both protection reliability and generation quality

**Weaknesses:**

Here are the weaknesses of the paper:
- The criticism to perturbation-based methods is overstated, like in the following sentence: "...unrealistic assumption that all images in the training dataset of malicious users are pre-perturbed by the protector". The authors should argument why they consider this assumption unrealistic.
- the following scenario lacks realism: "We propose a more practical scenario: protectors are typically large AI companies that provide
pre-trained generative models or offer personalization services directly to downstream users."
- the experimental validation is limited: the current approach is compared only against two other methods. Only one personalization approach, represented by Dreambooth, is presented.

**Questions:**

Besides the weaknesses mentioned above, here are my other concerns that should be addressed:
- the first and most concerning one is that the current paper relies heavily on the previous BadT2I approach (Zhai et al., 2023): it considers exactly three types of different attacks, but with different objectives (targets). Therefore, the authors should criticallly discuss how their approach is different from the BadT2I method and clearly state their scientific contributions.
- Section 3.2, lines 189-190: the authors make the following statement: "Recent studies have shown that T2I diffusion models are vulnerable to backdoor attacks, where adversaries controlling the training process can embed triggers to achieve malicious objectives." Please add references to support your claim.
- The evaluation against only two baselines (BadT2I and Personalization Shortcut) is insufficient. Additional comparison with other relevant approaches should be included (e.g. SIREN, EvilEdit)
- In order to validate the generality of your approach, you should validate it on other personalization approach, e.g. Textual Inversion, LoRA-based adaptation, etc.
- Rename section 4.3 'Discussion'. The goal of a 'Discussion' section is not to present additional experimental results, but to address several key aspects: to comment on the main findings, explain the insights they provide, relate them to existing work, acknowledge the limitations of the current approach, etc.

---

> ### Author Response · Authors · 2025-11-25
>
> Thank you for the feedback and spending your time reviewing our paper. Please find below responses to your comments and the changes we made in the revision (in magenta color).
>
> > **W1: Limitations of Perturbation-Based Protections**
>
> As stated in **Line 52** of the introduction, downstream training datasets in real-world scenarios typically contain a heterogeneous mixture of data sources, such as original unperturbed versions of the protected images, user-captured photos, or newly synthesized content, which the protector cannot control. This naturally breaks the assumption that every training sample is perturbed beforehand. Moreover, as illustrated in **Figure 1**, perturbations are applied only once prior to training, and protectors lack any control over the subsequent data processing steps. Even minor transformations or preprocessing operations performed during training can easily diminish or neutralize the perturbations, substantially reducing their intended protective effect.
>
> In addition, existing perturbation-based methods mainly focus on degrading the quality or fidelity of generated images. This degradation does not guarantee that sensitive visual attributes are fully concealed, leaving residual semantic cues that may still leak protected content. We have incorporated these clarifications into the revised manuscript to more clearly articulate why such assumptions do not hold in realistic adversarial training settings.
>
> > **W2: Practicality of the Protection Scenario**
>
> We clarify that our intention is not to claim that all real-world deployments strictly follow this scenario, but rather that it is more realistic than the assumptions underlying perturbation-based defenses, which typically require every image in a downstream training dataset to be pre-perturbed by the protector—an assumption that is difficult to satisfy in practice.
>
> In contrast, our scenario aligns with the increasingly common ecosystem in which large AI companies provide foundation models, APIs, or personalization/fine-tuning services directly to downstream users. For example, companies such as OpenAI, Google, and Stability AI offer model-as-a-service frameworks where users upload images for customization, embedding, or fine-tuning tasks. In such settings, protectors (the model providers) naturally have control over the model weights and system interface but do not control the user’s image collection pipeline. This structure makes model-level protection mechanisms more feasible and deployable compared to input-perturbation approaches.
>
> > **W3 & Q4: On Scalability Beyond DreamBooth**
>
> In **Sec. 4.4** of the revised version (in **teal color**), we have included experiments on **LoRA**, **SDXL**, and **Textual Inversion** (take target-backdoor as examples), as shown in **Table 8**. The results indicate that **PersGuard** maintains high effectiveness on **LoRA** and **SDXL**, performing comparably to the full **DreamBooth** fine-tuning reported in the initial submission. Regarding **Textual Inversion**, we observed a performance drop compared to weight-tuning methods. We attribute this to the inherent mechanism of **TI**, which restricts updates to the text embedding space rather than model weights, making the protection easier to override during inversion. We have added a detailed discussion on these architectural distinctions and acknowledged optimizing protection for embedding-only methods as a valuable direction for future work.
>
> > **Q1: Contribution**
>
> We acknowledge that our method shares similarities with **BadT2I**, but there are two key differences. First, our focus is on **personalization protection**, rather than addressing backdoor threats directly. Second, we demonstrate that existing **BadT2I** methods fail in personalization scenarios because backdoors are easily removed through personalized fine-tuning. To address this, we propose an additional **Backdoor Retention Loss** to ensure that backdoors are not discarded during user personalization, improving the robustness of our approach.
>
> > **Q2: Lack of Reference**
>
> We have now added references to recent works in the revision. These references are included in **Section 3.2** and the bibliography.
>
> > **Q3: Extra Baselines**
>
> We have added **EvilEdit** to our experiments in **Table 1**. Similar to **BadT2I**, we observe that backdoors are removed during the fine-tuning process. Regarding **SIREN**, we note that it is primarily a passive defense based on validation, which differs from the active defense approaches used in our work and other baselines. Therefore, a direct comparison is not feasible, but we have cited **SIREN** in the related work for completeness.
>
> > **Q5: Rename Section 4.3 'Discussion'**
>
> Thank you for the valuable suggestion. In the revised version, we have updated the section’s title and adjusted the wording accordingly.

---

> > ### Comment · Reviewer_5pjA · 2025-11-27
> > **Official Comment by Reviewer 5pjA**
> >
> > After reading the authors' rebuttal, I could say that my concerns have been partially addressed. The approach presented in the paper is largely incremental, and does not meet the typical level of novelty and rigor expected for a venue like ICLR. Therefore, I maintain my initial rating which is 4.

---

> > > ### Author Response · Authors · 2025-11-27
> > >
> > > Thank you again for your time and for revisiting our rebuttal. We fully understand and respect your decision.
> > >
> > > We just want to briefly clarify that, although our experiments compare with several prior methods (BadT2I, EvilEdit, etc.), which may give the impression that our method is an incremental extension of one of them, this is not our design intention. These baselines are compared as reference for readers and reviewers, not as the starting point of contribution.
> > >
> > > Our core idea is to repurpose backdoors as an active personalization protection mechanism at the model level, which is fundamentally different from (1) existing perturbation-based protections that operate on input images, and (2) prior diffusion backdoor attacks whose goal is malicious manipulation rather than defensive control under personalization. We appreciate your candid assessment and will further sharpen the positioning and technical depth in future versions.

---

### Official Review · Reviewer_rL9w · 2025-10-26

**Soundness:** 2
**Presentation:** 2
**Contribution:** 2
**Rating:** 2
**Confidence:** 5

**Summary:**

This paper introduces a novel defense mechanism against unauthorized personalization and fine-tuning attacks (i.e. DreamBooth misuse) on text-to-image diffusion models. Instead of relying on fragile input perturbations, PersGuard embeds a protective backdoor directly into the model so that when an attacker fine-tunes on protected content, the backdoor is automatically triggered to erase, alter, or watermark outputs. The method jointly optimizes three objectives—Backdoor Behavior (ensuring trigger activation), Prior Preservation (maintaining normal generation), and Backdoor Retention (preventing forgetting during downstream fine-tuning). Experiments were conducted on Stable Diffusion 2.1 across DreamBooth and CelebA-HQ datasets. PersGuard outperforms prior perturbation-based defenses (Anti-DB, PAP, SimAC, DisDiff) and backdoor baselines (BadT2I, Personalization Shortcut), maintaining protection even when only one protected sample is present.

**Strengths:**

1. The design of $L_{BR}$ is clever, maintaining the robustness of the inplanted backdoor by moving the parameters of the model towards the direction where the backdoor can be erased, so as to minimize the gradient during the post-tuning phases conducted by the malicious user.

2. The writing is good, and the paper is easy to follow.

**Weaknesses:**

1. **Inappropriate baselines examined**: The paper mainly compares PersGuard with perturbation-based approaches like Anti-Dreambooth, which are originally designed for protecting the published image from being unauthorizedly utilized by the attacker. While the purpose of PerGuard is significantly different, it aims to make the released model unable to be used for personalization fine-tuning. The abilities of the protectors are thereby different as well. In PersGuard, the protector apparently has much stronger capabilities to modify the model weights, while the perturbation-based methods cannot, making the comparison not only unfair but very likely meaningless. The comparison should be conducted between the model-fine-tuning-based methods, such as IMMA [1], Meta-Unlearning [2], and ResAlign [3].

2. **Confusing experiment demonstration**: The white-box setting, being the most fundamental one though, is NOT an appropriate setting to show the contribution of PersGuard. The assumption that the protector knows the placeholder (i.e., 'sks') that the attacker would use is too naive, trivial, and impractical. The experimental results under gray-box and black-box settings should be highlighted to better demonstrate the robustness of the proposed method against unseen identifiers and prompts. I thereby suggest that the authors revise the main result part to include more results in these two settings.

3. **Lack of backdoor capacity demonstration**: Only up to 3 backdoors are implanted into the model, but usually, many more concepts need to be protected in the real-world setting. The discussion on the scalability of the method is necessary and would enhance the contribution of this paper.

4. **No scalability discussion**: Only Dreambooth on SD 2.1 was examined. I strongly suggest that the author test more personalization methods, such as Textual-Inversion and LoRA-based Dreambooth.

In summary, I would believe that although this paper may have proposed a novel solution, the paper itself is still in a very immature state, especially for its problematic experiment design.

***Reference***

[1] Zheng, Amber Yijia, and Raymond A. Yeh. "Imma: Immunizing text-to-image models against malicious adaptation." European Conference on Computer Vision. Cham: Springer Nature Switzerland, 2024.

[2] Gao, Hongcheng, et al. "Meta-unlearning on diffusion models: Preventing relearning unlearned concepts." Proceedings of the IEEE/CVF International Conference on Computer Vision. 2025.

[3] Li, Boheng, et al. "Towards Resilient Safety-driven Unlearning for Diffusion Models against Downstream Fine-tuning." The Annual Conference on Neural Information Processing Systems (NeurIPS), 2025

**Questions:**

See in weakness.

---

> ### Author Response · Authors · 2025-11-25
>
> Thank you for the feedback and spending your time reviewing our paper. Please find below responses to your comments and the changes we made in the revision (in teal color).
>
> > **W1: Inappropriate Baselines Examined**
>
> We thank the reviewer for pointing out these relevant works. We have carefully studied them and added a comprehensive comparison in the revised **Appendix J**.
>
> | Method                | **Designed for Personalization** | **Multilayer Optimization** | DINO$_c$($\downarrow$) | CLIP$_c$($\downarrow$) |
> |-----------------------|----------------------------------|----------------------------|------------------------|-------------------------|
> | Normal                | -                                | -                          | 0.8311                 | 0.2932                  |
> | IMMA | $\checkmark$                         | $\checkmark$               | 0.7245                 | 0.2863                  |
> | ESD  | $\times$                         | $\times$                   | 0.7812                 | 0.2916                  |
> | SDD | $\times$                         | $\times$                   | 0.7797                 | 0.2948                  |
> | Meta-Unlearning | $\times$                         | $\checkmark$               | 0.6447                 | 0.2844                  |
> | PersGuard (Ours)      | $\checkmark$                     | $\times$                   | **0.3449**             | **0.2334**              |
>
> 1. **Comparison with IMMA**: We confirm IMMA is a relevant baseline as it shares the goal of preventing personalization. However, as shown in the table, **PersGuard** significantly outperforms IMMA. This performance gap is primarily due to IMMA's reliance on bi-level optimization, which is prone to instability and convergence to sub-optimal protection solutions. In contrast, **PersGuard** utilizes an efficient single-level optimization approach, ensuring superior and consistent effectiveness.
>
> 2. **Comparison with Concept Erasure Methods (ESD, SDD, Meta-Unlearning)**: We extended our evaluation to these methods despite their different design goals. The results confirm they are ineffective for our task. The goal misalignment is key: these methods are designed to erase existing concepts, whereas our goal is to prevent the learning of new, non-existent personalization concepts. Applying erasure objectives in our context is therefore conceptually flawed and practically ineffective. Furthermore, **Meta-Unlearning**, like IMMA, incurs high computational costs due to its bi-level formulation.
>
> 3. **Note on ResAlign**: We were unable to provide a comparison as the official code is not publicly available.
>
> > **W2: Confusing Experiment Demonstration**
>
> We appreciate the reviewer's suggestion to emphasize practical robustness. We have revised the paper to include more comprehensive results under gray-box and black-box settings in the main text (see **Table 6**).
> While we agree the white-box setting is idealized, it remains the standard benchmark adopted by state-of-the-art methods [1-4]. Retaining this setting is crucial for a fair and direct performance comparison against prior works, which primarily focus on white-box scenarios and rarely address black-box settings. As suggested, we now highlight the gray- and black-box results. The experiments demonstrate that **PersGuard** consistently outperforms perturbation-based baselines in these challenging scenarios. Unlike previous methods, **PersGuard-UID** does not rely on specific user training images or known placeholders, showing superior generalization against unseen identifiers and prompts.
>
>
> ---
>
> [1] Van Le, Thanh, et al. "Anti-dreambooth: Protecting users from personalized text-to-image synthesis." Proceedings of the IEEE/CVF International Conference on Computer Vision. 2023.
> [2] Liu, Yisu, et al. "Disrupting diffusion: Token-level attention erasure attack against diffusion-based customization." Proceedings of the 32nd ACM International Conference on Multimedia. 2024.
> [3] Wang, Feifei, et al. "Simac: A simple anti-customization method for protecting face privacy against text-to-image synthesis of diffusion models." Proceedings of the IEEE/CVF Conference on Computer Vision and Pattern Recognition. 2024.
> [4] Ye, Xiaoyu, et al. "Duaw: Data-free universal adversarial watermark against stable diffusion customization." arXiv preprint arXiv:2308.09889 (2023).

---

> ### Author Response · Authors · 2025-11-25
>
> > **W3: Lack of Backdoor Capacity Demonstration**
>
> We have expanded our study with additional experiments evaluating larger backdoor capacities, covering both (1) multiple face identity images within the same category ('person') and (2) multiple backdoors across different categories (see **Appendix E.2**). The results show that for identities within the same category, protection effectiveness decreases slightly as capacity increases, which is expected given the growing intra-class complexity. In contrast, when backdoors are assigned to different categories, the effectiveness remains consistently strong, indicating minimal cross-interference. These findings demonstrate that **PersGuard** can support substantially larger protection sets in practice.
>
> > **W4: On Scalability Beyond DreamBooth**
>
> In **Sec. 4.4** of the revised version, we have included experiments on **LoRA**, **SDXL**, and **Textual Inversion** (take target-backdoor as examples), as shown in **Table 8**. The results indicate that **PersGuard** maintains high effectiveness on **LoRA** and **SDXL**, performing comparably to the full **DreamBooth** fine-tuning reported in the initial submission. Regarding **Textual Inversion**, we observed a performance drop compared to weight-tuning methods. We attribute this to the inherent mechanism of **TI**, which restricts updates to the text embedding space rather than model weights, making the protection easier to override during inversion. We have added a detailed discussion on these architectural distinctions and acknowledged optimizing protection for embedding-only methods as a valuable direction for future work.
>
> > **W5: Experimental Design of Paper**
>
> Thanks for recognizing the novelty of our proposed solution and for the suggestions regarding the experimental design. In the revised version and the appendix, we have incorporated all of the experiments you suggested to better support and validate the effectiveness of our method.

---

### Official Review · Reviewer_GDi4 · 2025-11-01

**Soundness:** 2
**Presentation:** 2
**Contribution:** 2
**Rating:** 4
**Confidence:** 3

**Summary:**

This paper presents a defense-based backdoor framework to prevent unauthorized T2I personalization. Unlike the existing methods which fail when clean or transformed images are mixed in the training data, the proposed method embeds the backdoor into the model, so that the backdoors activate only when the prompts contain the identifiers. The proposed method introduce three loss functions to achieve the unified optimization objective. The results show that the proposed method outperforms the existing baselines.

**Strengths:**

1. First thing which makes the difference is that, existing methods assume full control over training data. Whereas, this method  assumes the protector controls only the model, which better aligns with the real-world applications.

2. The method remains effective even when attackers mix protected images with clean or augmented data that is a known failure for existing perturbation-based methods.

3. Among three introduced loss functions, backdoor retention loss seems more meaningful, because backdoor erosion while fine-tuning is a common issue in practice.

4. Includes white-box, gray-box, multi-identity, and facial privacy experiments, with LLM-based metrics and ablation studies.

5. Protected models maintain FID and DINO scores close to clean models on general and class-specific prompts.

**Weaknesses:**

1. The paper claims to be “the first to introduce a novel backdoor-based protection approach” (Sec 1). However, backdoor attacks for personalization methods like DreamBooth have already been studied. For example, [1] demonstrated that personalization itself can serve as a “shortcut” for few-shot backdoor injection, and other works have explored backdoors in T2I models via textual triggers or encoder manipulation [2,3]. While PersGuard repurposes backdoors for defense, it builds directly on frameworks like BadT2I (Sec 3.3). Thus, the novelty lies in the defensive application and retention mechanism, not in the concept of backdooring T2I personalization.

2. The target-backdoor replaces a protected human subject with an unrelated object such as “Superman”. This violates the fundamental expectation of personalization that prompts yield semantically coherent outputs consistent with the subject category. If the target is too distant i.e., person → rabbit, the output becomes nonsensical, revealing the defense and failing to respect the true notion of T2I personalization.

3. I think the prior preservation is coming from dreambooth. if its correct, while its reuse is justified, it should not be claimed as a novel contribution.

4. Experiments are conducted on SD 2.1, while the more latest SD variants such as SD 3, and SD3.5 are publicly available. The authors do not justify this choice. Also, their is a significant architectural difference between the SD2.1 and the newer SD variants. The older ones are based on UNET, whereas, the newer ones are DiT-based models.

[1] Personalization as a Shortcut for Few-Shot Backdoor Attack against Text-to-Image Diffusion Models
[2] Injecting Bias in Text-To-Image Models via Composite-Trigger Backdoors
[3] BAGM: A Backdoor Attack for Manipulating Text-to-Image Generative Models

5. [Minor] Though, DAAM is a popular choice for attention maps extraction, it aggregates attention heuristically across layers and timesteps. Directly hooking attentions from cross-attention blocks can give more correct maps.

**Questions:**

1. How does PersGuard fundamentally differ from these prior efforts beyond the defensive objective?
2. The target-backdoor replaces a protected subject (e.g., a person) with a semantically unrelated object (e.g., “Superman”). Doesn’t this violate the core expectation of personalization that outputs remain coherent with the subject category? How do you reconcile this with the goal of “preserving utility” for legitimate users?
3. The prior preservation loss appears identical to the one used in DreamBooth. Is this component truly a contribution of your work, or is it borrowed directly?
4. The backdoor retention loss is a key innovation. Can you provide ablation results showing how much fine-tuning (e.g., number of steps, learning rate) the backdoor can withstand before degrading?
5. Why was Stable Diffusion 2.1 chosen for all experiments, given that SD 3/3.5 are now standard? Have you evaluated PersGuard on more modern DiT-based architectures, and if not, what are the anticipated challenges?
6. In gray-box settings, the paper shows that universal prompt training improves robustness. But what about black-box scenarios where the attacker uses completely unseen identifiers or prompt templates? Is PersGuard fundamentally limited to white/gray-box assumptions?

---

> ### Author Response · Authors · 2025-11-25
>
> Thank you for the feedback and spending your time reviewing our paper. Please find below responses to your comments and the changes we made in the revision (in red color).
>
> > **W1 & W3 & Q1 & Q3: Novelty Claim**
>
> For **W1 & Q1**, we would like to clarify a misunderstanding. As stated in our paper, we claim to be the first to introduce a backdoor-based protection approach, and we do not claim to be the first to study backdoor attacks in personalization tasks. As you mentioned, prior works such as [1] and [3] have explored backdoor attacks, and we have already discussed and compared them with our work in the paper. Our method differs in that **PersGuard** uses backdoors specifically for defensive protection, rather than for launching attacks. We have also added citation [2] in the revised version to further clarify this distinction.
>
> For **W3 & Q3**, your understanding is correct. However, in our stated contributions in the introduction, we do not highlight prior preservation as our contribution; instead, our work focuses on proposing three backdoor objectives and developing a unified framework that incorporates these three losses. We have clarified this point and updated the introduction and method sections accordingly in the revised version.
>
> > **W2 & Q2: Expose Defense**
>
> Our goal is to prevent malicious personalization while still allowing valid personalization. As shown in **Fig. 3**, previous perturbation-based methods all degrade personalization results, so this trade-off is necessary for effective protection. We agree that extreme target shifts (e.g., person → rabbit) may lead to nonsensical outputs. However, our approach focuses on preventing malicious attacks, not all personalization. As shown in **Table 2**, our method maintains high performance on benign tasks and does not expose the defense, ensuring both effectiveness and stealth.
>
> > **W4 & Q5: Model Version**
>
> We have now included experiments in **Table 7** with other models, including the updated SD variants (SD-1.5, SD-2, SD-3, and SD-3.5). We would like to emphasize that our method is designed to be generalizable and independent of specific architectures or versions, as our training objective is end-to-end.
>
> > **W5: Additional Cross-Attention Visualizations**
>
> Following your advice, we have added additional visualizations obtained by directly hooking the cross-attention blocks of the multimodal models. These results are now included in **Table 11** in the Appendix.
>
> > **Q4: Robustness of Backdoor Retention Under Fine-Tuning**
>
> In the initial version of **Appendix Sec. B**, we already report the training and fine-tuning hyperparameters, including the number of steps and learning rate mentioned in the question. Moreover, the previous versions of **Fig. 5** and **Fig. 6** in the Appendix show how loss and evaluation metrics evolve as the fine-tuning steps increase. From these curves, we can observe that the backdoor retention loss prevents the backdoor from being overwritten from the very beginning of fine-tuning and that the protection effect is maintained throughout up to 50 fine-tuning steps.
>
> To further explore the impact of excessive fine-tuning on the backdoor, we also show in **Figure 7** of the appendix the changes observed during 500 steps of fine-tuning. We observe that, during the first 200 steps, the unprotected image quickly undergoes personalization, while the protected image maintains the target class output due to the backdoor. Although the backdoor begins to be gradually overwritten after 200 steps, we find that beyond this point, the model becomes overfitted to the personalized target, losing the ability to generate diverse and effective images.
>
> > **Q6: Black-box Scenario**
>
> As discussed in **Section 3.2**, we clarify that our universal prompt training is specifically motivated by and designed to address the challenge of unseen identifiers and prompt templates (the gray-box setting). The black-box scenario you describe is indeed the most challenging, as the protector additionally lacks access to the user's specific training images. To address this stringent case, we have added new black-box experiments where the user's images differ entirely from those used by the protector (see the revised **Table 6**).
>
> Results confirm that, unlike perturbation-based defenses that heavily rely on known or controlled input images, **PersGuard** maintains strong and consistent protection even when the attacker uses unseen inputs. This empirically demonstrates that **PersGuard** generalizes effectively against fully unconstrained attacks, validating its robustness beyond gray-box assumptions.
>
> ---
>
> [1] Personalization as a Shortcut for Few-Shot Backdoor Attack against Text-to-Image Diffusion Models
> [2] Injecting Bias in Text-To-Image Models via Composite-Trigger Backdoors
> [3] BAGM: A Backdoor Attack for Manipulating Text-to-Image Generative Models

---

### Official Review · Reviewer_DbBB · 2025-11-02

**Soundness:** 2
**Presentation:** 2
**Contribution:** 2
**Rating:** 4
**Confidence:** 3

**Summary:**

This work introduces the novel idea of utilizing backdoor attack techniques for personalization protection — PersGuard, which injects protective backdoor triggers into released diffusion models. Specifically, the authors add the 'backdoor retention loss' to maintain the backdoor capabilities from downstream fine-tuning tasks.

**Strengths:**

1. The idea of utilizing the backdoor attack techniques is novel to me.
2. The authors provide several settings to validate the PersGuard's capability, including gray-box settings, multi-objective projection, and facial identity protection.
3. Especially, they test whether the single trigger (e.g., *sks*) would apply to different protected objects, which would adapt to more general privacy cases.

**Weaknesses:**

1. **Unclear scenarios of gray-box settings.** In gray-box cases, the *universal identifier tokens* and *universal training prompts* are unclear to me. Do you coverage more identifier tokens during the protector stage (as described in Figure 1)? If so, the PersGuard still requires a significant number of identifier tokens (for personalization training), which sounds impractical.
2. **Concerns about comparisons with Perturbation-Based Protections.** In Figure 3 (b), what are the prompts of multimodal LLM to get the Protection Success Rate on the $y$-axis?
3. Follow 2. and your method to convert the personalization identifier to other results. The stealthy of the model looks weak because the personalization fine-tuned models would generate the results far away from the prompt. In contrast, perturbation-based protection methods might develop a more semantically preserved prompt. What do you think?
4. Follow 2., beyond the ablation study of PersGuard on gray-box settings, I suggest the authors should provide perturbation-based protections for different settings for a clearer status of the existing personalization protection methods.
5. **Concerns about the core questions of utilizing BadT2I for this scenario.** While the authors design the 'backdoor retention loss' to maintain the backdoor capabilities, the empirical result in Table 5 shows that the $\mathcal{L}_\mathcal{BR}$ provides minor improvement. Since the ideas of *Backdoor Behavior Loss* and *Prior Preservation Loss* originate from BadT2I, I have some concerns about the core questions the authors address.

**Questions:**

1. After reviewing the idea of PersGuard, I conjecture that PersGuard cannot remove the existing concept that has already been trained in the pre-trained diffusion model. For example, a public figure such as an actor or a politician. While I know the scope of this work might not coverage this scenario, is that possible to extend this idea in this direction? If so, I think the potential of this direction would be more substantial.

---

> ### Author Response · Authors · 2025-11-25
>
> Thank you for the feedback and spending your time reviewing our paper. Please find below responses to your comments and the changes we make in the revision (blue color).
>
> >**W1: Unclear scenarios of gray-box settings.**
>
> We have made more explicit scenario settings Sec 4.4 in the revision and Sec. G in appendix.
> For PersGuard-UI, in the protector stage we use a pool of identifier tokens (e.g., “sks”, “abc”, “[A*]”, etc.) combined with several generic class names (e.g., “dog”, “animal”, “pet”). For PersGuard-UP, we employ multiple universal training prompts (e.g., “This is an image of …”, “The photo depicts …”, “A portrait of …”).
>
> However, in our experiments we only use 10 identifier tokens/class names and 5 training prompts, and this modest set already demonstrates strong universality to unseen identifiers and prompts. This indicates that the universal protection strategy is practically feasible and does not require exhaustive coverage of all potential attacker choices. We reckon that this effectiveness arises because the identifier tokens, class names, and prompts used by the protector are semantically related to those an attacker would likely choose (e.g., “animal” vs. “pet”, or “This is an image of …” vs. “A portrait of …”), enabling protection to generalize across their shared semantic neighborhoods. We believe these observations point toward promising directions for developing even more efficient gray-box protection schemes in future work.
>
> > **W2: Concerns about comparisons with Perturbation-Based Protections**
>
> In the experiment of Fig.3, we employed five prompts to query each LLMs:
> + Do you think these two images are of the same class?
> + Are these two images belonging to the same category?
> + Do these images depict the same type of object or scene?
> + Would you classify these two images under the same label?
> + Is the semantic content of these two images similar enough to be considered the same class?
>
> We have now added these details to section H of appendix to facilitate better reproducibility for readers.
>
> > **W3: Stealthiness of the Model**
>
> We respectfully have a different opinion that PersGuard exhibits strong stealthiness. As shown in Tables 1 and 2, for general tasks and non-protected personalized generations, our protected model behaves almost identically to the clean model, meaning benign users will not perceive any difference.
>
> On the other hand, as illustrated in Fig. 3, both perturbation-based defenses and our backdoor-based PersGuard inevitably lead to unsatisfactory outputs once protection is triggered. Generating low-quality (previous methods) or incorrect-category images (PersGuard) for protected identifiers is the intended protective behavior and is orthogonal to stealthiness for normal usage. This is analogous to standard NSFW defenses that degrade or refuse unsafe generations without reducing model stealthiness for benign users.
>
> > **W4: Concerns about comparisons with Perturbation-Based Protections**
>
> As suggested, we have expanded our evaluation to include representative perturbation methods under both gray-box and black-box assumptions (see the revised Table 6).
>
> Our results indicate that perturbation-based defenses maintain reasonable effectiveness in the gray-box setting. However, their performance dramatically drops in the stricter black-box setting, which involves completely unseen identifiers and prompts. In contrast, our method, PersGuard, is designed to be agnostic to the attacker's specific identifier and training images, demonstrating robust and consistent effectiveness across all settings, especially under the challenging black-box scenario.
>
> > **W5: Concerns about backdoor retention loss**
>
> As shown in Table 5, the Backdoor Retention Loss ($L_{BR}$) is essential. Without it, as in the second row of Table 5, where only $L_{BB}$ and $L_{PP}$ are used, $DINO_b$ is very low, and $DINO_c$ is high, indicating that personalization results are not protected and the backdoor target class is not generated.
>
> We also emphasize the key differences between our approach and **BadT2I**:
> 1. Our focus is on *personalization protection*, not revealing general backdoor threats.
> 2. We show that **BadT2I** fails in personalization scenarios because backdoors are easily removed through fine-tuning, which we address with the Backdoor Retention Loss.
>
> > **Q1:Extension to Pre-trained Concepts**
>
> Thanks for the insightful idea. We acknowledge that the current PersGuard cannot remove existing concepts already embedded in personalized diffusion models.  However, AI companies, the protectors, can proactively minimize further risk or privacy leakage by iteratively updating model versions and abolish the unprotected versions.

---

### Author Response · Authors · 2025-12-01
**Summary for AC: ICLR 2026 Submission 7372 - PersGuard**

Dear PCs, SACs, ACs, and Reviewers,

Thank you very much for your valuable contributions to our work. To assist the newly assigned AC and help reduce their workload, we provide below a summary of the key points from the reviews and the reviewer-author discussions.

## Strengths
We are grateful that the reviewers overall viewed our problem setting and methodology as meaningful and technically sound. In particular:

- **Novel backdoor-based protection approach**: The idea of introducing a novel backdoor-based protection approach (PersGuard) for preventing unauthorized personalization. (All reviewers)
- **Unified objective with backdoor retention**: Unified objective incorporating backdoor retention for robustness under fine-tuning. (GDi4, rL9w, 5pjA)
- **Broad evaluation**: Comprehensive evaluation across threat models and tasks. (All reviewers)
- **Utility and stealth preservation**: Preservation of normal utility and stealth for benign users. (GDi4, DbBB)
- **Clarity and organization**: Clarity of writing and organization. (rL9w, 5pjA)

## Concerns and Our Addressing
Overall, although the reviewers raised some weaknesses and concerns, it can be observed that the majority of them are concentrated on aspects such as experimental design, experimental coverage, and baseline selection, rather than the correctness or feasibility of the core method. In response, we have supplemented corresponding experiments and analyses in both the revised main text and appendix, and explained them point by point in the rebuttal. The following provides a more detailed summary by theme.

- **Concerns about novelty and contributions** (GDi4: Weaknesses 1,3; 5pjA: Questions 1,3)
  **Our Addressing**: We clarified that our core contribution lies in the defensive repurposing of backdoor techniques for personalization protection, introducing the backdoor retention mechanism as a novel addition to counter erosion during fine-tuning. We clearly distinguished our work from attack-oriented studies (e.g., BadT2I, EvilEdit) and added relevant citations while updating the methods section. Although Reviewer 5pjA considers our contribution incremental, we reckon this perception stems from our extensive comparisons with numerous backdoor or perturbation-based methods in the experiments; however, these comparisons are solely intended to clearly highlight the limitations of existing approaches.

- **Concerns about experimental design and scalability** (rL9w: Weaknesses 1-5; GDi4: Weaknesses 4, Questions 4-6; 5pjA: Weaknesses 3, Questions 3,4; DbBB: Weaknesses 4)
  **Our Addressing**: We added comparisons with IMMA, Meta-Unlearning, ESD, SDD, EvilEdit (Appendix J, Tables 1,6,7); experiments on newer models (SD 1.5/2/3/3.5, Table 7), methods (LoRA, SDXL, Textual Inversion, Table 8), larger backdoor capacities (Appendix E.2), and extended fine-tuning ablations (Figs. 5-7, Appendix B). We emphasized gray/black-box results in the main text (Table 6) and confirmed robustness with unseen images/prompts. We justified baselines as focused on personalization protection, not concept erasure.

- **Concerns about assumptions and realism** (DbBB: Weaknesses 1-3; 5pjA: Weaknesses 1,2; GDi4: Weaknesses 2, Questions 2)
  **Our Addressing**: We detailed gray-box settings (Sec 4.4, Appendix G) and added LLM prompts for reproducibility (Appendix H). We clarified perturbation limitations (e.g., heterogeneous data breaks assumptions) and defended stealthiness/utility preservation (Tables 1,2; degradation necessary for protection). We justified the scenario as more practical than perturbations, aligning with AI company ecosystems. We added direct attention visualizations (Table 11, Appendix).

- **Concerns about backdoor retention and extensions** (DbBB: Weaknesses 5, Questions 1; GDi4: Questions 1)
  **Our Addressing**: We emphasized retention's essential role (Table 5) and differences from BadT2I. We acknowledged limitations for pre-trained concepts but suggested iterative model updates as a proactive extension.
----
Because we substantially expanded the experiments and appendix during the discussion period in response to reviewer feedback, our rebuttal and the revised version were submitted relatively late. Consequently, some reviewers may not have had sufficient time to carefully read the full rebuttal and the newly added experimental results, and may therefore not have updated their scores or continued the discussion accordingly. Nevertheless, we have made every effort to address the reviewers’ concerns point-by-point.

Above, we have faithfully summarized all reviewer comments and our corresponding responses, hoping that this will assist the AC's work. We are deeply grateful to the reviewers, AC, SAC, and PC, for their dedicated effort and excellent work. Their insightful feedback has further strengthened our paper. The authors offer their sincere respect and appreciation to all involved!

Sincerely,
Authors

---

### Note · Authors · 2025-12-18

I have read and agree with the venue's withdrawal policy on behalf of myself and my co-authors.